# Mufu: Multilingual Fused Learning for Low-Resource Translation with LLM

**Zheng Wei Lim**[♡][*] **Nitish Gupta**[◇] **Honglin Yu**[◇] **Trevor Cohn**[♡,◇]

[♡]The University of Melbourne  [◇]Google

## Abstract

Multilingual large language models (LLMs) are great translators, but this is largely limited to high-resource languages. For many LLMs, translating in and out of low-resource languages remains a challenging task. To maximize data efficiency in this low-resource setting, we introduce Mufu, which includes a selection of automatically generated multilingual candidates and an instruction to correct inaccurate translations in the prompt. Mufu prompts turn a translation task into a postediting one, and seek to harness the LLM's reasoning capability with auxiliary translation candidates, from which the model is required to assess the input quality, align the semantics cross-lingually, copy from relevant inputs and override instances that are incorrect. Our experiments on En-XX translations over the Flores-200 dataset show LLMs finetuned against Mufu-style prompts are robust to poor quality auxiliary translation candidates, achieving performance superior to NLLB 1.3B distilled model in 64% of low- and very-low-resource language pairs. We then distill these models to reduce inference cost, while maintaining on average 3.1 chrF improvement over finetune-only baseline in low-resource translations.

## 1 Introduction

The most advanced of large language models (LLM) have demonstrated remarkable competence in translation-related tasks (Robinson et al., 2023; Hendy et al., 2023; Alves et al., 2024; Kocmi & Federmann, 2023; Raunak et al., 2023), but lag behind in translations involving lower-resource languages (Robinson et al., 2023; Hendy et al., 2023; Zhu et al., 2024; Lu et al., 2024), compared to specialized neural machine translation (NMT) systems like NLLB (Costa-jussà et al., 2022). This performance gap is caused primarily by scant pre-training data in these languages (Wei et al., 2023; Yuan et al., 2024; Alves et al., 2024), and is difficult to overcome despite growing efforts to support translations of long-tail languages (Kudugunta et al., 2024; Bapna et al., 2022; Lu et al., 2024).

In this work, we introduce multilingual fused learning (Mufu), which combines multilingual context and a postediting task when translating into lower-resource languages using LLMs.[1] Mufu-style prompts (see Table 1, top block) include several multilingual translation candidates along with a postediting target, from which a model learns "in-context" to translate from languages with which the target language is more closely aligned due to cultural relevance, geographical and genealogical proximity. We rely on a larger, more competent multilingual teacher model to generate auxiliary translations in these languages, which help disambiguate inputs and improve cross-lingual semantic alignment in a translation task. Given a task to postedit, LLMs are capable of "translating" better by iteratively improving the fluency and naturalness of the translation candidates (Chen et al., 2023).

The goal is to induce in LLMs multi-step reasoning akin to chain-of-thought (CoT) (Wei et al., 2022), as the models are required to assess the input quality, align the candidates cross-lingually, and improve the final translation by drawing from the correct input and overriding incorrect instances. Translating this way can be challenging for small models with limited reasoning capacity. Inspired by Wang et al. (2023), we further propose finetuning against Mufu prompts, which allows the models to learn how to best exploit and benefit from the multilingual context.

---

[*]Work done during an internship at Google.

[1]We borrow the name from 幕府 (mù fǔ), a secretariat for the imperial Chinese officers dating back to 229 BC (Wikipedia contributors, 2024).

---

**0** The English sentence has been translated into Malay, Javanese, Sundanese, Indonesian, Minangkabau and Achinese. These translations may contain errors. Correct the translation from English to Achinese.

**1** English: The proposed amendment already passed both houses in 2011.
**2** Automatic Malay: Pindaan yang dicadangkan telah diluluskan oleh kedua-dua dewan pada tahun 2011.
**3** Automatic Javanese: Amandemen sing diusulake wis ditampa dening loro omah ing taun 2011.
**4** Automatic Sundanese: Amandemen anu diusulkeun parantos lulus duanana imah dina 2011.
**5** Automatic Indonesian: Amandemen yang diusulkan sudah disahkan oleh kedua majelis pada tahun 2011.
**6** Automatic Minangkabau: Amandemen nan diusulkan alah disetujui dewan legislatif pado taun 2011.
**7** Automatic Achinese: Amandemen nyang geupeugah nyan ka geupeugot bak keu-2 bak thôn 2011.
**8** Corrected Achinese:

---

Reference: Amandemen nyang geuusong ka geuteurimoeng lé banduwa majeulis bak thôn 2011.

---

Baseline instruction: Translate from English to Achinese.

---

Table 1: Prompt template for `mufu5` (top block) with Achinese as an example, which includes an instruction (line 0), an input (line 1, blue), five multilingual candidates (lines 2-6, orange) and a postediting target (line 7, red). For baseline we omit lines 2-7, replacing *Corrected Achinese* with *Achinese* and the initial instruction with the baseline instruction in purple. In `postediting`, we remove auxiliary languages (teal) in the instruction along with the multilingual candidates, retaining only the postediting target.

We show that the best Mufu model, finetuned only with hundreds of parallel examples in each language pair, is competitive against the teacher model and the benchmark NLLB 1.3B distilled model, scoring on average 2.7 higher chrF on FLORES-200 devtest and 0.7 on NTREX test sets in En-XX translations.[2] Importantly, Mufu works well on a range of pre-trained models including PaLM2 and Gemma, despite limited data and the fact that Gemma models are English-centric models that have not been trained for multilingual capabilities (Anil et al., 2023; Gemma Team et al., 2024). Our experiments further demonstrate knowledge distillation on Mufu models to be effective in reducing the inference cost, while maintaining competitive advantage against benchmark.

## 2 MULTILINGUAL FUSED LEARNING

### 2.1 COMBINING TWO LEARNING PARADIGMS

Few-shot in-context learning (ICL) is incredibly effective for eliciting translations from an LLM (Winata et al., 2021; Lin et al., 2022), but is usually less performant than more compute- and data-intensive finetuned models (Zhang et al., 2023b; Vilar et al., 2023; Xu et al., 2024; Lu et al., 2024). On one hand, ICL improves translations of LLMs by allowing for informative contexts that induce reasoning processes in the model, and prompt the model to reach a latent feature space that is otherwise difficult to access with shorter input (Wei et al., 2022; Wang et al., 2023; Vilar et al., 2023; Puduppully et al., 2023; Zhu et al., 2024; Zhang et al., 2023a). On the other hand, LLMs produce higher quality final predictions with parameter tuning. Motivated by Wang et al. (2023), our work combines the strengths of both learning paradigms by finetuning LLMs with reference output against multilingual prompts, and substantially improves the overall quality of LLMs' translations over finetuned-only models, under a low-data condition.

### 2.2 MAXIMIZING DATA EFFICIENCY WITH MULTILINGUAL AUXILIARY TRANSLATIONS

Beyond providing few-shot examplars in a translation prompt, we incorporate translations in other languages as auxiliary information to the task. Learning to translate this way facilitates semantic alignment beyond the lexical level, by allowing the encoding of rich knowledge network embedded in the multilingual translations. This multilingual context includes a draft translation in the target language, thus turning the difficult task of translating from scratch into a postediting task. Taken together, this approach can be considered similar to CoT rationales, as we expect LLM to be able to disambiguate words and align across multilingual context, to copy from high-quality inputs and to disregard instances that are less informative or are of poor quality. Unlike typical CoT, however, Mufu models do not predict the chain of thought and is instead provided as a rich context for intermediate reasoning in translation.

---

[2]Based on the performance of PaLM2 XXS–NTL (mufu20), further details in Section 3.3.

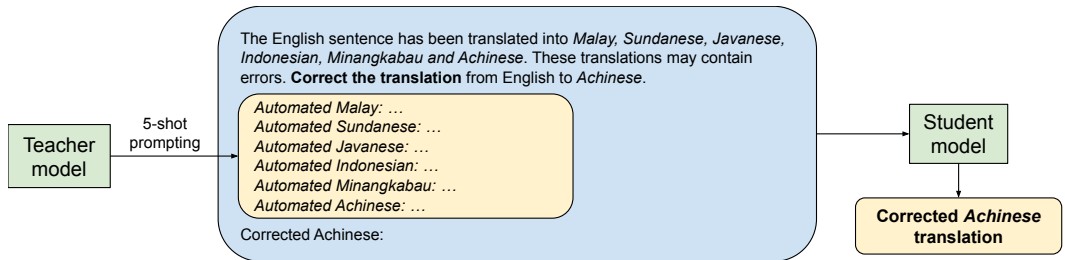

Figure 1: Mufu involves two iterations. First, a teacher model generates a set of multilingual auxiliary translations and a postediting target. These translations then become part of the input during the second iteration, where the student model learns in-context to produce the corrected target translation. We then finetune the student model against target references.

In practice, to obtain and to incorporate the auxiliary translations and postediting target in context, Mufu requires two iterations. During the first iteration, a teacher model is required to generate the intermediary translations. These translations are later included as part of the input for a student model, which learns in-context to correct the target translation in the second iteration.[3] We illustrate an example of this process in Figure 1, where the teacher model first translates the same input from English to auxiliary translations in Malay, Sundanese, Javanese, Indonesian, Minangkabau and Achinese (the target language).[4] These outputs are then added as part of the in-context prompt for the student model, along with an instruction to correct the target translation.

## 3 EXPERIMENTS

### 3.1 DATA AND EVALUATION

As a low-data setup, we train and validate on the FLORES-200 dev split (Costa-jussà et al., 2022), which differs from the usual practice of reserving the split entirely for validation.[5] Out of 997 source sentences in the split, we randomly sampled 787 sentences as the train set, 100 sentences as the validation data, and another 100 sentences to perform initial prompt selection. We reserve the remaining ten source sentences, from which we sample five-shot exemplars used in generating auxiliary translations in the first iteration. Each of the source sentences is paired with translations in 203 languages, from which we finetune the student models to translate from English into a subset of 201 target languages.[6] Some languages use more than one writing systems—for example, Achinese can be written in Latin and Arabic scripts; we treat translations into different scripts as individual language pairs.

We evaluate our approach using chrF, a character overlap statistic (Popović, 2015). The finetuned models are tested on FLORES-200 devtest split for the ideal in-domain setting where train and test conditions are closely matched. The source sentences of FLORES-200 are sampled from Wikipedia—to assess our finetuned models out of domain, we use NTREX (Federmann et al., 2022), which comprises translations of English news data, on which we evaluate 112 languages, the subset of languages also found in FLORES-200.[7]

### 3.2 PROMPT STYLE AND AUXILIARY LANGUAGES

We test a variety of prompts with a one-shot prompting and choose an instruction that list all auxiliary languages (e.g., *... from English to Malay, Sundanese, Javanese, ...*) over an instruction for the model to infer these languages from the prompt (e.g., *... from English to several languages as specified*). We also prepend *Automatic/Corrected* labels to the language tags in the auxiliary translations instead

---

[3]The student may be the same model as the teacher in this setup.

[4]See Section 3.2 for details on how the intermediate languages are chosen.

[5]As described in Costa-jussà et al. (2022).

[6]The two languages omitted are Akan and Twi.

[7]The languages from FLORES-200 not supported in NTREX are shown as dashed entries in Table 8 (Appendix A.5).

of *Candidate/Reference* pair. We show in Table 1 an example template of a Mufu instruction, in contrast with the `baseline` setup where we provide only an instruction to translate in the prompt, without any multilingual context or postediting target. Further details on prompt selection can be found in Appendix A.1.

To select the most relevant auxiliary languages in Mufu, we rely on language data from URIEL (Littell et al., 2017) to select the closest languages by geological and genetic distance (equally weighted) for each target language, and arrange them by the farthest to closest in the prompt. Several languages are not included in the URIEL repository, in which case we sampled their auxiliary languages randomly.[8] For the full list of auxiliary languages used in Mufu prompts, see Appendix A.2.

We finetune with Mufu prompt over a varying number of auxiliary translations: `postediting` (`mufu0`) contains only a postediting target and does not include any multilingual context; `mufu`$N$ incorporates $N \in \{5, 10, 20\}$ auxiliary multilingual translations in addition to a postediting target.

### 3.3 MODELS

The teacher model, PaLM2 S (also known as Bison), has shown excellent multilingual and translation capability (Anil et al., 2023), but there remains a significant performance gap between higher-resource and lower-resource languages—we report the teacher performance in Section 4 and show the gap can be largely reduced by the student models through Mufu. During the first iteration, the teacher model generates auxiliary translations for each instance with 5-shot prompting. For all prompt setups described in the previous section, we perform supervised finetuning jointly over 201 languages for En-XX translation over a range of student models: PaLM2 XXS (Gecko), PaLM2 XS (Otter), Gemma 2B-IT and Gemma 7B-IT; given the same auxiliary translations generated previously.

When comparing the performance across student models, it is worth noting that PaLM2 are multilingual LLMs with superior initial translation capacity compared to Gemma models, which have not received any specialized training on multilingual tasks (Gemma Team et al., 2024). We also further pre-train PaLM2 XXS, the smallest model from PaLM2 family, on a corpora derived from the Next-Thousand-Language (NTL) effort, which comprise monolingual and parallel sentences in 1000+ languages (Caswell et al., 2020; Bapna et al., 2022). We refer to this version of the model as PaLM2 XXS–NTL henceforth.

## 4 RESULTS

We evaluate primarily using chrF rather than BLEU (Papineni et al., 2002), which heavily relies on tokenization that is underdeveloped for many low-resource languages.[9] Table 2 shows the mean chrF across 201 En-XX language pairs of all teacher, student and benchmark models; and Win%, the percentage of language pairs where the model outperforms a benchmark. NLLB models only support 198 of these language pairs—to facilitate comparison, we therefore report also the average chrF and win percentages over just these languages.[10]

When tested with in-domain FLORES devtest data, Mufu finetuned models gain substantially over their baselines. Turning a translation task to a postediting one is advantageous to the output quality, and we see further improvements with multilingual context in Mufu prompts. Mufu models also show superior performance compared to the teacher, with PaLM2 XXS–NTL exceeding teacher performance in 54.2% translation pairs respectively. The exception is regular PaLM2 XXS, which score better than the baseline but underperforms compared to the teacher and the smaller NLLB model, presumably due to its limited capacity.

In theory, it is possible for the student to be at least as good as the teacher through word-for-word copying from the postediting target. However, some Mufu translations are worse than the teacher.

---

[8]The languages not found in URIEL include Latgalian, Swahili, Kongo, Kanuri, Kanuri in Arabic script, Silesian, Pashto, Oromo, Guarani, Kabuverdianu, Tumbuka, Kimbundu, Filipino, Friulian, Dinka, Mongolian, Azerbaijani, Fulfulde, South Levantine Arabic, Uzbek, Sardinian, Limburgan, Persian, Tamazight, Crimean Tatar in Latin script, Dzongkha, Lombard and Dari.

[9]Nonetheless, we report the corresponding results in BLEU scores in Appendix A.4, which largely corroborate our main findings.

[10]The languages not supported by NLLB are Minangkabau in Arabic script, Arabic in Latin script and Santali.

| | | FLORES-200 devtest | | | | | NTREX | | |
|---|---|---|---|---|---|---|---|---|---|
| | | chrF ↑ (n=201) | chrF ↑ (n=198) | Win% vs. teacher | Win% vs. NLLB 1.3B | Win% vs. NLLB 54B | chrF ↑ (n=112) | Win% vs. teacher | Win% vs. NLLB 1.3B |
| PaLM2 S (teacher) | | 43.3 | 43.7 | - | 58.1 | 43.2 | 48.6 | - | 73.2 |
| NLLB 1.3B distilled | | - | 46.0 | 41.3 | - | 4.0 | 48.1 | 26.8 | - |
| NLLB 54B MoE | | - | 48.9 | 56.2 | 96.0 | - | - | - | - |
| PaLM2 XXS –NTL | baseline | 39.2 | 39.4 | 32.8 | 11.6 | 8.0 | 36.3 | 8.9 | 0.9 |
| | postedit | 42.5 | 42.8 | 34.8 | 19.2 | 10.6 | 40.6 | 9.8 | 3.6 |
| | mufu5 | 47.1 | 47.3 | 46.8 | 57.1 | 24.6 | 46.5 | 17.0 | 21.4 |
| | mufu10 | 48.0 | 48.3 | 52.2 | 75.3 | 32.7 | 47.7 | 17.0 | 35.7 |
| | mufu20 | **48.4** | **48.7** | 54.2 | 76.8 | 39.7 | 48.8 | 20.5 | 61.6 |
| | mufu5hrl | 42.9 | 43.1 | 34.3 | 20.7 | 10.6 | 41.0 | 10.7 | 3.6 |
| | mufu5tr | 44.4 | 44.6 | 42.3 | 33.8 | 19.1 | 43.0 | 11.6 | 7.1 |
| | mufu20+5hrl | 47.1 | 47.4 | 47.3 | 63.1 | 23.1 | 46.9 | 15.2 | 25.9 |
| | distilled | 45.1 | 45.5 | 42.8 | 35.4 | 17.1 | **49.0** | 45.5 | 48.2 |
| PaLM2 XXS | baseline | 35.8 | 35.9 | 26.9 | 7.6 | 5.5 | 34.2 | 5.4 | 1.8 |
| | postedit | 41.7 | 42.0 | 28.9 | 22.2 | 9.0 | **43.4** | 6.2 | 8.9 |
| | mufu5 | **41.9** | **42.2** | 30.8 | 20.2 | 11.6 | 43.1 | 7.1 | 8.9 |
| | mufu10 | 41.0 | 41.1 | 30.8 | 14.1 | 9.0 | 40.2 | 8.0 | 4.5 |
| | mufu20 | 41.1 | 41.2 | 30.8 | 14.1 | 9.5 | 40.3 | 8.0 | 4.5 |
| PaLM2 XS | baseline | 31.7 | 31.9 | 21.9 | 2.5 | 1.0 | 31.3 | 5.4 | 0.0 |
| | postedit | 43.8 | 44.1 | 36.8 | 28.3 | 16.6 | 43.3 | 8.9 | 10.7 |
| | mufu5 | 44.5 | 44.6 | 40.8 | 33.8 | 17.6 | 43.6 | 8.9 | 11.6 |
| | mufu10 | 44.5 | 44.7 | 40.3 | 36.9 | 19.1e | 43.6 | 9.8 | 13.4 |
| | mufu20 | **44.7** | **44.8** | 43.3 | 36.9 | 19.1 | **43.8** | 9.8 | 13.4 |
| PaLM2 S | baseline | 32.9 | 33.0 | 27.4 | 4.5 | 2.5 | 30.7 | 7.1 | 0.0 |
| | mufu20 | 47.0 | 47.1 | 51.2 | 58.6 | 27.6 | 45.6 | 17.9 | 26.8 |
| | mufu20lora | **47.2** | **47.5** | 99.0 | 72.2 | 59.8 | **50.1** | 91.1 | 83.9 |
| Gemma 2B | baseline | 34.4 | 34.4 | 28.9 | 9.1 | 4.0 | 29.2 | 6.2 | 0.9 |
| | postedit | 44.1 | 44.3 | 32.8 | 37.9 | 16.1 | 41.4 | 8.0 | 7.1 |
| | mufu5 | 45.1 | 45.3 | 37.8 | 49.5 | 22.1 | 43.2 | 9.8 | 9.8 |
| | mufu10 | 45.4 | 45.5 | 39.3 | 47.0 | 21.1 | 43.3 | 9.8 | 10.7 |
| | mufu20 | **45.5** | **45.6** | 39.3 | 47.5 | 22.6 | **43.6** | 10.7 | 13.4 |
| Gemma 7B | baseline | 39.9 | 40.0 | 33.3 | 15.7 | 9.5 | 35.1 | 7.1 | 0.9 |
| | postedit | 46.3 | 46.5 | 41.8 | 54.0 | 24.6 | 43.2 | 9.8 | 12.5 |
| | mufu5 | 47.2 | 47.3 | 49.3 | 60.6 | 27.6 | 43.4 | 9.8 | 11.6 |
| | mufu10 | 47.2 | 47.3 | 49.3 | 61.6 | 27.1 | 43.2 | 9.8 | 14.3 |
| | mufu20 | 47.6 | 47.7 | 51.7 | 63.6 | 29.6 | 43.6 | 11.6 | 17.9 |
| | mufu5hrl | 46.4 | 46.6 | 42.8 | 52.0 | 26.1 | 43.2 | 10.7 | 13.4 |
| | mufu5tr | 42.9 | 42.9 | 42.3 | 28.8 | 17.6 | 37.5 | 9.8 | 4.5 |
| | mufu20+5hrl | **47.7** | **47.8** | 51.2 | 66.7 | 30.7 | 44.1 | 12.5 | 17.9 |
| | distilled | 44.4 | 44.5 | 41.3 | 26.8 | 18.1 | **47.2** | 33.9 | 41.1 |

Table 2: Mean chrF scores and win percentages against PaLM2 S as teacher model for 201 En-XX language pairs; NLLB 1.3B distilled model and NLLB 54B MoE model for 198 language pairs. **Bold** values are the best chrF scores in a given model class. Red values are win rates above 50%. Mufu{5, 10, 20} indicate the number of non-target multilingual candidates in the prompt. We also report the distillation performance of PaLM2 XXS–NTL and Gemma 7B finetuned with mufu20.

We attribute this phenomenon to the limited amount of supervision in each language pair and autoregressive modeling objective with gold-standard translation—a strategy known to be inferior to distilling from model outputs (Kim & Rush, 2016; Wang et al., 2021; Finkelstein & Freitag, 2023). Mufu is effective for under-resourced languages with low-quality postediting candidates. However, improving high-quality translations in high-resource languages is harder and requires the student model to also learn the subtle differences between model- and human-generated output (Sizov et al., 2024; Zhang et al., 2024; Kocmi et al., 2024). It is also possible that the teacher model surpasses humans for some translations in high-resource languages—in which case, learning from the human translations could be detrimental.

Compared to NLLB 1.3B distilled, PaLM2 XXS–NTL finetuned with mufu20 translates better in nearly 77% language pairs. The best Mufu models also outperform NLLB 54B MoE in up to nearly

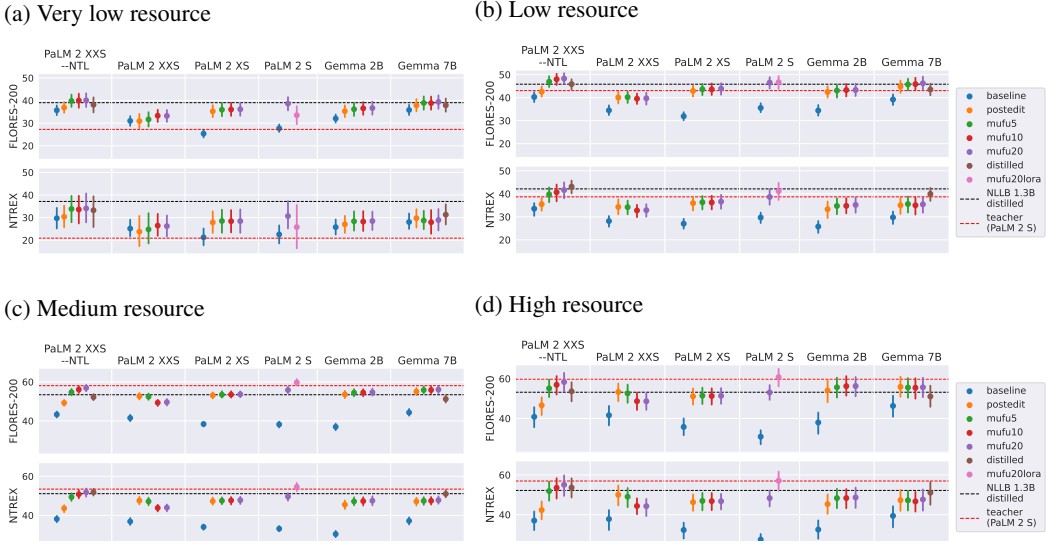

Figure 2: Mean chrF across languages of the same resource level. Mufu outperforms the baseline consistently, and improves upon translations by the teacher model in low and very-low resource languages. Mufu is also competitive against NLLB 1.3B distilled in translating into very low resource languages, and consistently outperforms the latter in low, medium and high resource setting. Note that the scales of y-axes are different for the top and bottom rows. Error bars shown are 95% confidence intervals across the language pairs.

40% of the translation pairs, despite being an order of magnitude smaller than the benchmark model. The result thus suggests the potential advantage in using higher-quality multilingual candidates produced by NLLB for Mufu.[11]

While we expect a decline in performance due to distribution shift when translating out-of-domain sentences of NTREX, Mufu models hold up well in comparison with the baseline. Most Mufu models no longer outperform the teacher model and NLLB 1.3B distilled, but PaLM2 XXS–NTL with mufu20 maintains an advantage over the NLLB model, scoring higher on average with 48.8 chrF, and is better in 62% language pairs.

The full results of PaLM2 XXS–NTL (mufu20) and Gemma 7B (mufu20) are reported in Appendix A.5. To generalize the performance of Mufu beyond PaLM2 and Gemma models, we additionally report the translation results of finetuned BLOOMZ 1B7 (Muennighoff et al., 2023) in Appendix A.6, which show significant improvement over baseline and postedit only conditions.

## 4.1 Performance in low-resource languages

Figure 2 shows the mean chrF of Mufu models in four language categories: very-low resource ($n = 68$), low resource ($n = 45$), medium resource ($n = 68$) and high resource ($n = 17$) languages.[12] Again, we compare against the teacher and NLLB 1.3B distilled models, indicated by the red and black dashed lines respectively.

We are most interested in the very-low-resource languages, where we observe all Mufu models obtain substantial gains over the teacher model. This shows Mufu is capable of overcoming noisy auxiliary candidates, since most low-resource target languages are in proximity with other low-resource languages, as included in the prompt. The best Mufu models are also competitive against NLLB 1.3B distilled, and maintain these advantages in low-, medium- and high-resource settings.

---

[11]We also extract translations from PaLM2 XXS–NTL by five-shot prompting (without any parameter updates), and find the translation quality to be worse than baseline finetuning, supporting Zhang et al. (2023b).

[12]The resource levels of each language were based on our subjective judgements on the accessibility of data and the competency of current translation systems to and from English. We report the resource levels of the languages in Appendix A.5, Table 8.

| | | FLORES-200 devtest | | | NTREX | |
|---|---|---|---|---|---|---|
| | | teacher | NLLB 1.3B | NLLB 54B | teacher | NLLB 1.3B |
| PaLM2 XXS –NTL | baseline | 56.9 | 16.8 | 11.4 | 32.3 | 0.0 |
| | postedit | 60.3 | 23.9 | 13.2 | 35.5 | 3.2 |
| | mufu5 | 78.4 | 56.6 | 28.9 | 54.8 | 19.4 |
| | mufu10 | 85.3 | 65.5 | 35.1 | 54.8 | 12.9 |
| | mufu20 | 85.3 | 63.7 | 36.0 | 64.5 | 38.7 |
| | distilled | 73.3 | 40.7 | 21.1 | 77.4 | 41.9 |
| Gemma 7B | baseline | 57.8 | 23.9 | 14.9 | 25.8 | 0.0 |
| | postedit | 71.6 | 42.5 | 27.2 | 25.8 | 6.5 |
| | mufu5 | 81.0 | 50.4 | 30.7 | 25.8 | 6.5 |
| | mufu10 | 81.9 | 51.3 | 29.8 | 22.6 | 6.5 |
| | mufu20 | 84.5 | 53.1 | 33.3 | 29.0 | 6.5 |
| | distilled | 71.6 | 33.6 | 26.3 | 61.3 | 25.8 |

Table 3: Win percentages measured over the 113 low and very-low resource languages for models shown in rows against, as columns, the teacher model, NLLB 1.3B distilled and NLLB 54B MoE. Win rates above 50% are in red.

In medium- and high-resource languages, Mufu models improve the most relative to the baseline, but fall short compared to the teacher model.

The win percentages of the best Mufu models, PaLM2 XXS–NTL and Gemma 7B, against the teacher model and NLLB models in low and very-low resource languages are reported in Table 3, which largely corroborate the results in Figure 2. Mufu models outperform the teacher in 78–85% of these languages on FLORES devtest and up to 64.5% on NTREX. Among the Mufu models, PaLM2 XXS–NTL is the most consistent, outscoring NLLB 1.3B in 64% and 39% languages. It is also impressive that the Mufu model beats NLLB 54B MoE in more than one third of the languages on FLORES devtest, given the substantial difference in training and capacity.

## 4.2 Cross-lingual alignment with attention and the effect of auxiliary translations in closely related languages

We present cross-lingual attention alignment of the finetuned models across Mufu input as a mechanistic explanation of the improvement in translation performance. Table 4 compares the translations by Gemma 2B finetuned with mufu5 prompt and the baseline prompt. *Tenth* is translated as *Keupulôh* by mufu5, which is close in form to the reference (*kesiploh*) and is untranslated in the postediting target and skipped entirely by the baseline model. The top block highlights parts of the input attended by the mufu-finetuned model, immediately before the production of *Keupulôh*, indicating transfer of the form from these auxiliary translations. The model also fixates on *Achinese*, the target language in this example.

Beyond outright copying, Mufu models are also capable of transliterating and translating from attention-aligned input that are dissimilar in form. Transliteration from Latin to Arabic script is observed in Achinese—an example where the model transliterates *Jamaika* into the correct Arabic form جامايكا, a word unseen in the postediting target and the baseline translation, is shown in Table 10 in Appendix A.7; whereas the translation of *minimum* to Mizo involves attention to Bengali, which differs from Mizo in form and script, as shown in Table 11.

We provide quantitative evidence in Figure 3, showing the sum of mean multi-head attention of all layers directed to different parts of mufu5 inputs from the generated candidate (normalized by length), across validation examples of a sample of language pairs. Apart from the postediting target, Indonesian auxiliary input is the most useful when translating into Achinese in both Latin and Arabic script; Myanmar receives the most attention relative to the other auxiliary inputs during the translation into Mizo; auxiliary translation in Rundi is helpful to the translations into Kinyarwanda, as Zulu is to Swati—some of these auxiliary translations receive comparable attention to the English source during the process.

| | The English sentence has been translated into Malay, Sundanese, Javanese, Indonesian, Minangkabau and Achinese. These translations may contain errors. Correct the translation from English to Achinese. |
|---|---|
| | English: In an ambush east of Bardia, the British captured the Italian Tenth Army's Engineer-in-Chief, General Lastucci.

Automatic Malay: Dalam satu serangan hendap di timur Bardia, British berjaya menangkap Ketua Jurutera Tentera Itali, Jeneral Lastucci.

Automatic Javanese: Ing serangan ing sisih wétan Bardia, Inggris nyekel Insinyur-ing-Kepala Tentara Italia Sepuluh, Jenderal Lastucci.

Automatic Sundanese: Dina hiji tewak di wétan Bardia, Inggris néwak Insinyur-in-Chief Tentara Italia, Jenderal Lastucci.

Automatic Indonesian: Dalam sebuah penyergapan di sebelah timur Bardia, Inggris menangkap Insinyur-in-Chief Angkatan Darat Italia Kesepuluh, Jenderal Lastucci.

Automatic Minangkabau: Dalam suatu penyergapan di timur Bardia, Inggris manawan Insinyur Kapalo dari Tentara Italia ka-10, Jenderal Lastucci.

Automatic Achinese: Bak sèngkeu bak timu Bardia, ureueng Inggeris geupeunan ureueng Italia Tenth Army's Engineer-in-Chief, General Lastucci.

Corrected Achinese:Lam seubap senyeurôh di sebelah timu Bardia, Inggreh neukapol roh Insinyur-in-Chief Angkatan Darek Italia |
| mufu5 | Lam seubap senyeurôh di sebelah timu Bardia, Inggreh neukapol roh Insinyur-in-Chief Angkatan Darek Italia **Keupulôh**, Jeneral Lastucci. |
| baseline | Bak saboh sembuh kira-kira Bardia, Ureueng Inggreh ipeumeunangan Enreng Italia Jumat Pkat Teuntra-dalam-Cahya, Jendral Musoh Lekka. |
| reference | Lam penyerangan di timu Bardia, ureueng Inggréh geudrop pangulëë insinyur angkatan darat **kesiploh** Italia, Jenderal Lastucci. |

Table 4: Translations from English to Achinese. The word *Tenth* in English is untranslated in the postediting target and baseline, but is translated into *Keupulôh* (cf. *kesiploh* in reference) by Gemma 2B finetuned with mufu5 prompt. The highlighted text shows the aligned attention across mufu5 prompt right before the production of *Keupulôh*, indicating form transfer from the multilingual input (*Sepuluh* in Javanese, *Kesepuluh* in Indonesian, *ka-10* in Minangkabau). Note that the attention presented here is the mean value across multiple heads and layers. Tokens with aggregated attention values under .01, .06, .13, .22 are colored in white, light gray, dark gray and black respectively.

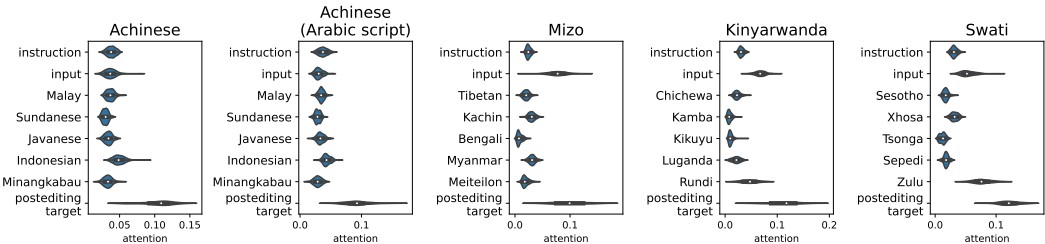

Figure 3: Sum of self-attention from the tokens of generated candidate (e.g., "Corrected Achinese: ...") to the instruction, input, auxiliary translations and the postediting target. Some auxiliary translations receive more attention than the input (e.g., Indonesian vs. input in translating into Achinese in Latin and Arabic scripts; Zulu vs. input in translating into Swati). Note that a significant portion of attention is placed at the generated sequence itself, which is omitted from the plot.

### 4.3 ABLATION

**Mufu iteratively improves translations where teacher and student are the same model**. We report the results of finetuned PaLM2 S (baseline and mufu20) in Table 2 and Figure 2 to demonstrate the efficacy of Mufu in setups where the student and teacher are the same model.

**Mufu mitigates overfitting**. PaLM2 XS and PaLM2 S finetuned with the baseline method overfit and perform worse than PaLM2 XXS (Table 2).[13] Mufu is largely resistant to the problem, showing consistent improvement with increasing model size. To further reduce overfitting, we experiment with LoRA finetuning ($r = 16$) (Hu et al., 2022) on PaLM2 S with mufu20 (`mufu20lora`). This setup

---

[13]It is possible that the models overfit to translations in high-resource languages, but not in low-resource languages. Thus, a reasonable approach would be to terminate high-resource-language training early (i.e., as a form of curriculum learning). We leave this experiment to future work.

pushes the model's win rates to 99% and 91% in FLORES-200 test and NTREX; and leads to better performance than NLLB 54B in nearly 60% translation directions (Table 2). Figure 2a, however, reveals that mufu20 with LoRA, while being highly resistant to overfitting with few parameter updates, is less effective than full finetuning on very-low-resource languages. The result is presumably related to recent findings that LoRA with low-rank perturbation underperforms compared to full finetuning in newly acquired skills (lower-resource languages), but forgets less of the prior knowledge gained during pre-training (higher-resource languages) (Biderman et al., 2024).

**Mufu works best with closely related auxiliary languages**. To test if Mufu is still effective without these careful selection of auxiliary languages, we additionally finetune PaLM2 XXS–NTL and Gemma 7B with mufu5 prompt consisting of only five high-resource languages chosen to simulate colonial influence: Dutch, Russian, French, Chinese and Spanish; and report the result in Table 2 (`mufu5hrl`).[14] While having less relevant multilingual context is better than having no context at all, the improvement is far below the model's upper threshold of translation capacity that we observe in the other Mufu variants. Adding these languages to mufu20 (`mufu20+5hrl`, Table 2) improves Gemma 7B's translations but undermines the performance of PaLM 2 XXS–NTL, as the model is distracted from highly informative candidates in relevant languages.[15]

**Mufu's performance is predominantly driven by multilingual candidates.** In `mufu5tr`, we remove the postediting target and instruct the model (PaLM2 XXS–NTL and Gemma 7B) to translate given the other auxiliary candidates. Table 2 shows mufu5tr to be better than the postediting task alone, but combining both conditions (mufu5) yields the best performance.

**Distilling Mufu models reduces inference cost and retains accuracy gains**. Translating with Mufu admittedly incurs a high inference cost given the need to generate auxiliary translations. Thus, we propose distilling Mufu models with the best performance in low-resource languages to reduce the cost to the baseline level (Kim & Rush, 2016). For distillation data, we use the 6193 English sentences from NLLB seed data (Costa-jussà et al., 2022), and sample 6000 English sentences from past WMT General Tasks test sets (2009–2018) that are not found in NTREX.[16] We use the simple sequence knowledge distillation method from Kim & Rush (2016), which involves supervised fine-tuning of the student model against teacher-predicted sequences.

We choose to distill PaLM2 XXS–NTL and Gemma 7B finetuned with mufu20 for their strong performance in low resource languages. The `distilled` models are competitive against baseline and the teacher model across all languages (Table 2), as well as in low-resource languages (Figure 2, Table 3). Given the mixture of domains in the distillation data, it is not surprising to see the distilled model outperforming the initial model in NTREX, in spite of the latter having never been exposed to gold translation output from the news domain. This signals strong potential to improve out-of-distribution performance of other Mufu models without additional parallel data source.

## 4.4 FAILURE CASES

Although translation quality improves in most languages pairs, there are a few cases where Mufu underperforms the baseline. One reason is the use of randomly sampled auxiliary languages for some target languages (Section 3.2). In practice, however, only four out of these 28 target languages has auxiliary languages that diverge sufficiently from the target languages and hurt the translation performance consistently.[17] Another major cause is the inclusion of auxiliary inputs of extremely poor quality—with three or more bad auxiliary translations, the input becomes more of a distraction than providing informative context. We provide an example of such input in Appendix A.8.

## 5 RELATED WORK

**ICL for translation.** Vilar et al. (2023), Zhang et al. (2023a) and Zhu et al. (2024) find exemplar quality plays a more important role than semantic relevance in prompting for good translations.

---

[14]Where the target language is one of these languages, we replace the auxiliary input with a translation candidate in Arabic.

[15]For target languages with high-resource languages also appearing in the related auxiliary languages, we include additional related languages such that there are 25 distinct auxiliary candidates in total in the context.

[16]https://github.com/facebookresearch/flores/blob/main/nllb_seed/README.md.

[17]The languages are Kanuri in Arabic script, Fulfulde, Tamazight and Kimbundu.

Few-shot ICL is however less effective in translating out of English than into English, contributing to the huge performance gap between low-resource and high-resource languages (Robinson et al., 2023; Zhu et al., 2024). Ghazvininejad et al. (2023) improve LLM's translation of rare words by providing multiple word-word hints derived from bilingual dictionaries. Mufu does not require bilingual dictionaries, which can be hard to obtain for very-low-resource languages; and has shown remarkable improvement over baselines when translating into low-resource languages, which are among the harder translation directions.

**Multilingual CoT reasoning for translation.** LLMs are capable of chain-of-thought reasoning with multilingual prompts (Shi et al., 2023; Chai et al., 2024). Zhu et al. (2024) find cross-lingual translation exemplars to improve translations from lower-resource languages to English. Puduppully et al. (2023) iteratively combines chunks of zero-shot translated input, assuming monotonicity between the source and target languages. He et al. (2024) translate with LLM using synthetic keyword pairs, input topics and semantically related exemplars extracted from the same model, but rely on quality estimators to select the final predictions.

**Low-resource translation with LLM.** Low-resource languages are notoriously difficult for LLMs. Claude Opus, an LLM nearly three orders of magnitude larger than Mufu models (Anthropic, 2024), outscores NLLB 54B in only 33% pairs of languages in the En-XX directions (Enis & Hopkins, 2024). This is in spite of the fact that the model showing signs of contamination from FLORES-200 (Enis & Hopkins, 2024). A growing body of work has nonetheless shown progress in the effort to reduce the translation performance gap across language pairs, as well as that between LLMs and supervised NMT models (Tanzer et al., 2024; Zhu et al., 2024; Bansal et al., 2024; Lu et al., 2024; Enis & Hopkins, 2024; Bapna et al., 2022; Hendy et al., 2023). LLMs are comparable to human in translations of unseen low-resource languages, when given the same language material (Tanzer et al., 2024; Reid et al., 2024). Bansal et al. (2024) augments an LLM with a smaller LLM of higher expertise in multilinguality to improve low-resource XX-En translation, adding only a small set of trainable parameters. Lu et al. (2024) extend the vocabulary of LLaMa models (Touvron et al., 2023; Dubey et al., 2024) and continually pre-train the models with large-scale monolingual, parallel and synthetic data involving 102 languages. The pretrained models are superior to M2M-100 (Fan et al., 2021) in En-XX translations, but are nevertheless outmatched by NLLB 1.3B, which is more advanced than M2M-100.

# 6 DISCUSSION

We present Mufu in this work, a method that maximizes data efficiency in low-resource translations with multilingual ICL and finetuning. Our analysis on cross-attention behaviour in Mufu-finetuned models provides evidence that the method extends LLM's capability in multilingual reasoning. That is, given any Mufu-style prompt, the finetuned models are capable of discerning input quality from multilingual candidates, aligning the input semantics across languages beyond orthographic similarity, and improving the candidate translation drawing only from informative context. Mufu models are stronger than the teacher model in low-resource languages and achieve consistent improvement over baseline finetuned models.

Mufu showcases a practical application of multilingual CoT to serve under-resourced languages, but the method carries two limitations. First, while it is largely robust against imperfect multilingual candidates, there seems to be a minimum quality threshold under which Mufu translates worse than the baseline. It would be, however, possible to extract higher-quality auxiliary translations from a stronger teacher (e.g., NLLB 54B), or to perform simple automated checks (e.g., for repetitions) to remove poor auxiliary candidates, to ensure the usefulness of the multilingual context. Second, relative to NMT models, Mufu incurs additional latency for improved accuracy, e.g., mufu5 improves the baseline model by 25.4% on average with six times more inference cost. The tradeoff is also evident in knowledge distillation on Mufu models with limited performance gains despite minimal latency. Thus it is up to the practitioners to train using a more comprehensive data set, or to consider the acceptable tradeoff in their use cases. There are nevertheless alternative LLM distillation methods that learn from model-generated text with substantial gains in generalization performance (Finkelstein & Freitag, 2023; Agarwal et al., 2024; Gu et al., 2024; Wang et al., 2024).

ACKNOWLEDGMENTS

Special thanks to Isaac Caswell, Colin Cherry, Aditi Chaudhary, Grace Chung, Xin Yu and Daniel Formosa for their helpful feedback on drafts of the paper and research support.

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

## A  APPENDIX

### A.1  PROMPT SELECTION

Prior to conducting experiments reported in the main text, we tested several versions of Mufu prompt on 100 sentences from FLORES-200 dev split reserved for prompt selection (see Section 3.1). We focused on a handful of target languages in the preliminary experiment: Achinese, Balinese, Buginese, Banjar and Minangkabau; using a fixed set of auxiliary languages: Indonesian, Malay, Javanese, Sundanese and Arabic. Auxiliary candidates for prompt selection were first generated by PaLM2 S via one-shot prompting:

> Translate from English to <target language>.
>
> English: Maybe one day, your great grandchildren will be standing atop an alien world wondering about their ancient ancestors?
> <target language>: <reference translation>
>
> English: <input>
> <target language>:

We then evaluated different versions of the prompt using the same model during the second iteration, where auxiliary candidates were included in the instruction similar to the template shown in Table 1 in the main text. We swapped out listed languages in the instruction with "... *several languages as specified*", and discovered it to be sub-par compared to the original prompt. We also experimented with prepending "*Candidate/Reference*" to the language tags in place of "*Automatic/Corrected*", and found the latter to yield superior performance. Note that these preliminary experiments on prompt variation do not involve finetuning, and we arrive at a final prompt template based on results derived entirely from zero-shot prompting.

## A.2 AUXILIARY LANGUAGES

Table 5 lists the custom set of auxiliary languages for each target language included in Mufu-style prompt. The languages are selected based on URIEL repository as described in Section 3.2, and are arranged from farthest to closest. Target languages assigned with random auxiliary languages are marked with †.

| Target language | Auxiliary languages |
| --- | --- |
| Achinese | Buginese, Samoan, Shan, Vietnamese, Malagasy, Ilocano, Myanmar (Burmese), Fijian, Maori, Sinhala, Lao, Khmer, Thai, Balinese, Banjar, Malay, Javanese, Sundanese, Indonesian, Minangkabau |
| Achinese in Arabic script | Buginese, Samoan, Shan, Vietnamese, Malagasy, Ilocano, Myanmar (Burmese), Fijian, Maori, Sinhala, Lao, Khmer, Thai, Balinese, Banjar, Malay, Javanese, Sundanese, Indonesian, Minangkabau |
| Afrikaans | Bemba (Zambia), Danish, Xhosa, Swedish, Sesotho, Norwegian, Chichewa, Faroese, Icelandic, Tswana, Shona, Yiddish, Swati, Tok Pisin, Luxembourgish, Zulu, Sepedi, German, Tsonga, Dutch |
| Albanian | Slovenian, French, Finnish, Romanian, Sicilian, Ewe, Basque, Italian, Croatian, Bengali, South Azerbaijani, Serbian, Hungarian, Egyptian Arabic, Bosnian, Amharic, Macedonian, German, Greek, Bulgarian |
| Amharic | Sango, Hausa, Kinyarwanda, Rundi, Luo, Kamba (Kenya), Tunisian Arabic, Luganda, Kikuyu, Maltese, Nuer, North Levantine Arabic, Mesopotamian Arabic, Najdi Arabic, Arabic, Hebrew, Egyptian Arabic, Somali, Ta'izzi-Adeni Arabic, Tigrinya |
| Arabic | Bulgarian, Turkish, Tamasheq, Somali, Hausa, Armenian, Georgian, Tunisian Arabic, Kurdish (Kurmanji), Amharic, Sorani Kurdish, Maltese, South Azerbaijani, Tigrinya, Ta'izzi-Adeni Arabic, Hebrew, Egyptian Arabic, North Levantine Arabic, Najdi Arabic, Mesopotamian Arabic |
| Arabic in Latin script | Bulgarian, Turkish, Tamasheq, Somali, Hausa, Armenian, Georgian, Tunisian Arabic, Kurdish (Kurmanji), Amharic, Sorani Kurdish, Maltese, South Azerbaijani, Tigrinya, Ta'izzi-Adeni Arabic, Hebrew, Egyptian Arabic, North Levantine Arabic, Najdi Arabic, Mesopotamian Arabic |
| Armenian | Romanian, Lithuanian, Turkmen, Kashmiri, Najdi Arabic, Icelandic, Turkish, Hindi, North Levantine Arabic, Irish, French, Mesopotamian Arabic, South Azerbaijani, German, Sorani Kurdish, Kurdish (Kurmanji), Georgian, Bengali, Greek, Bulgarian |
| Assamese | Marathi, Myanmar (Burmese), Sanskrit, Gujarati, Kachin, Sinhala, Mizo, Santali, Kashmiri, Bhojpuri, Tibetan, Magahi, Meiteilon (Manipuri), Awadhi, Punjabi, Hindi, Nepali, Maithili, Odia (Oriya), Bengali |
| Asturian | Luxembourgish, Romanian, German, Sicilian, Kabyle, Welsh, Irish, Haitian Creole, Esperanto, Italian, Venetian, Papiamento, Basque, Ligurian, Occitan, French, Catalan, Spanish, Galician, Portuguese |
| Awadhi | Meiteilon (Manipuri), Sindhi, Marathi, Tibetan, Sinhala, Sanskrit, Santali, Urdu, Assamese, Gujarati, Magahi, Kashmiri, Odia (Oriya), Bhojpuri, Bengali, Maithili, Punjabi, Chhattisgarhi, Nepali, Hindi |
| Ayacucho Quechua | Kabiyè, Finnish, Tamasheq, Basque, Mossi, Greek, Dyula, German, Bambara, Wolof, Bulgarian, Yiddish, Bengali, Haitian Creole, South Azerbaijani, Papiamento, Egyptian Arabic, Aymara, Amharic, Ewe |
| Aymara | Finnish, Hausa, Tamasheq, Basque, Mossi, Greek, Dyula, German, Bambara, Wolof, Bulgarian, Yiddish, Bengali, Haitian Creole, South Azerbaijani, Papiamento, Egyptian Arabic, Amharic, Ayacucho Quechua, Ewe |
| Azerbaijani† | Buginese, Cebuano, Chokwe, Icelandic, Fulfulde, Wolof, Norwegian, Luba-Lulua, Malayalam, Uyghur, Sorani Kurdish, Bambara, Myanmar (Burmese), Mandarin Chinese, Kabyle, Urdu, Tamazight, Zulu, German, Luxembourgish |
| Balinese | Vietnamese, Thai, Lao, Samoan, Khmer, Malagasy, Fijian, Maori, Pangasinan, Waray (Philippines), Ilocano, Cebuano, Buginese, Minangkabau, Malay, Javanese, Banjar, Achinese, Sundanese, Indonesian |
| Bambara | Kabyle, Finnish, Igbo, Basque, Greek, Yoruba, Fon, German, Bulgarian, Bengali, Kabiyè, Mossi, South Azerbaijani, Egyptian Arabic, Tamasheq, Amharic, Wolof, Hausa, Ewe, Dyula |
| Banjar | Thai, Vietnamese, Samoan, Lao, Malagasy, Khmer, Fijian, Maori, Pangasinan, Waray (Philippines), Ilocano, Cebuano, Minangkabau, Achinese, Buginese, Sundanese, Javanese, Balinese, Indonesian, Malay |
| Banjar in Arabic script | Thai, Vietnamese, Samoan, Lao, Malagasy, Khmer, Fijian, Maori, Pangasinan, Waray (Philippines), Ilocano, Cebuano, Minangkabau, Achinese, Buginese, Sundanese, Javanese, Balinese, Indonesian, Malay |
| Bashkir | Lithuanian, Latvian, German, Belarusian, Bulgarian, Bengali, Finnish, Egyptian Arabic, Amharic, Tajik, Georgian, Armenian, Turkish, Russian, Uyghur, Turkmen, South Azerbaijani, Kyrgyz, Kazakh, Tatar |
| Basque | Luxembourgish, Finnish, Ligurian, Esperanto, Ewe, Greek, Occitan, Irish, Galician, Bulgarian, Bengali, Catalan, South Azerbaijani, Asturian, Spanish, Egyptian Arabic, German, Portuguese, Amharic, French |

| Target language | Auxiliary languages |
| --- | --- |
| Belarusian | Greek, Danish, Macedonian, Swedish, Hungarian, Bosnian, Romanian, German, Estonian, Slovenian, Russian, Croatian, Serbian, Bulgarian, Slovak, Latvian, Lithuanian, Czech, Polish, Ukrainian |
| Bemba (Zambia) | Xhosa, Umbundu, Lingala, Sesotho, Swati, Afrikaans, Luo, Tswana, Tsonga, Chokwe, Luba-Lulua, Sepedi, Zulu, Kamba (Kenya), Rundi, Kikuyu, Luganda, Shona, Kinyarwanda, Chichewa |
| Bengali | Chhattisgarhi, Marathi, Gujarati, Myanmar (Burmese), Sanskrit, Tibetan, Sinhala, Punjabi, Meiteilon (Manipuri), Bhojpuri, Kashmiri, Mizo, Magahi, Santali, Awadhi, Hindi, Nepali, Maithili, Odia (Oriya), Assamese |
| Bhojpuri | Sindhi, Mizo, Meiteilon (Manipuri), Gujarati, Marathi, Sanskrit, Sinhala, Chhattisgarhi, Tibetan, Santali, Kashmiri, Punjabi, Assamese, Odia (Oriya), Hindi, Awadhi, Bengali, Nepali, Magahi, Maithili |
| Bosnian | Sicilian, Romanian, Lithuanian, Belarusian, Hungarian, German, Venetian, Polish, Russian, Italian, Greek, Ukrainian, Czech, Slovak, Albanian, Bulgarian, Slovenian, Macedonian, Serbian, Croatian |
| Buginese | Thai, Vietnamese, Lao, Khmer, Samoan, Malagasy, Minangkabau, Fijian, Maori, Pangasinan, Waray (Philippines), Malay, Achinese, Banjar, Ilocano, Cebuano, Balinese, Indonesian, Sundanese, Javanese |
| Bulgarian | Latvian, Venetian, German, Lithuanian, Turkish, Hungarian, Belarusian, Russian, Albanian, Romanian, Polish, Czech, Slovak, Greek, Slovenian, Ukrainian, Bosnian, Croatian, Serbian, Macedonian |
| Cantonese | German, Bulgarian, Waray (Philippines), Thai, South Azerbaijani, Egyptian Arabic, Amharic, Shan, Bengali, Khmer, Lao, Ilocano, Tibetan, Pangasinan, Vietnamese, Mizo, Meiteilon (Manipuri), Kachin, Myanmar (Burmese), Mandarin Chinese |
| Catalan | Bulgarian, Bengali, Romanian, Luxembourgish, German, Esperanto, Haitian Creole, Papiamento, Kabyle, Sicilian, Basque, Venetian, Italian, Galician, French, Asturian, Ligurian, Portuguese, Occitan, Spanish |
| Cebuano | Lao, Vietnamese, Samoan, Khmer, Minangkabau, Malagasy, Cantonese, Fijian, Maori, Malay, Achinese, Indonesian, Banjar, Balinese, Sundanese, Pangasinan, Buginese, Javanese, Ilocano, Waray (Philippines) |
| Chhattisgarhi | Mizo, Kannada, Santali, Sinhala, Punjabi, Kashmiri, Assamese, Urdu, Telugu, Bhojpuri, Maithili, Magahi, Gujarati, Nepali, Marathi, Bengali, Sanskrit, Odia (Oriya), Awadhi, Hindi |
| Chichewa | Lingala, Malagasy, Luo, Xhosa, Luba-Lulua, Sesotho, Chokwe, Afrikaans, Luganda, Tswana, Kamba (Kenya), Kikuyu, Rundi, Swati, Tsonga, Bemba (Zambia), Kinyarwanda, Sepedi, Shona, Zulu |
| Chokwe | Sesotho, Sango, Swati, Luo, Tsonga, Afrikaans, Kamba (Kenya), Tswana, Sepedi, Kikuyu, Bemba (Zambia), Zulu, Rundi, Luganda, Shona, Kinyarwanda, Chichewa, Lingala, Luba-Lulua, Umbundu |
| Crimean Tatar in Latin script[†] | Sanskrit, Fulfulde, Tamil, South Levantine Arabic, Sundanese, Limburgan, Azerbaijani, Guarani, Latvian, Kikuyu, Kinyarwanda, Irish, Tatar, Egyptian Arabic, Lingala, Hausa, Friulian, Maori, Tamazight, Oromo |
| Croatian | Latvian, Romanian, Lithuanian, Greek, Belarusian, Albanian, Italian, Russian, Venetian, Polish, Hungarian, German, Ukrainian, Bulgarian, Czech, Macedonian, Slovak, Serbian, Slovenian, Bosnian |
| Czech | Italian, Romanian, Dutch, Macedonian, Danish, Luxembourgish, Lithuanian, Venetian, Hungarian, Belarusian, Russian, Bosnian, Bulgarian, Serbian, Ukrainian, German, Croatian, Slovenian, Polish, Slovak |
| Danish | Greek, Scottish Gaelic, Bulgarian, Bengali, Norwegian Nynorsk, Yiddish, Lithuanian, Tok Pisin, Esperanto, Afrikaans, Polish, Czech, Faroese, French, Icelandic, Luxembourgish, German, Dutch, Swedish, Norwegian |
| Dari[†] | Japanese, Aymara, Pangasinan, Maltese, Ilocano, Turkmen, Faroese, Oromo, Igbo, Yoruba, South Levantine Arabic, Guarani, Kikuyu, Ayacucho Quechua, Lao, Balinese, Latvian, Fijian, Belarusian, Kabuverdianu |
| Dinka[†] | Umbundu, Uyghur, Arabic, Fijian, Catalan, Sorani Kurdish, Mandarin Chinese, Bulgarian, Bengali, Japanese, Ilocano, Spanish, Korean, Balinese, Kabuverdianu, Achinese, Tsonga, Macedonian, Friulian, Polish |
| Dutch | Bulgarian, Bengali, Ligurian, Occitan, Swedish, Scottish Gaelic, Faroese, Czech, Icelandic, Welsh, Yiddish, Tok Pisin, Irish, Esperanto, Afrikaans, Norwegian, French, Danish, German, Luxembourgish |
| Dyula | Finnish, Sango, Basque, Greek, Wolof, Igbo, German, Yoruba, Fon, Bulgarian, Bengali, South Azerbaijani, Egyptian Arabic, Hausa, Kabiyè, Amharic, Tamasheq, Mossi, Ewe, Bambara |
| Dzongkha[†] | Cantonese, Kashmiri, Fon, Aymara, Ayacucho Quechua, Albanian, Swati, Lingala, Ta'izzi-Adeni Arabic, South Levantine Arabic, Georgian, Italian, Norwegian Nynorsk, Crimean Tatar in Latin script, Kannada, Maltese, Fijian, Welsh, Shona, Igbo |
| Egyptian Arabic | Somali, South Azerbaijani, Albanian, Hausa, Kurdish (Kurmanji), Tigrinya, Amharic, Sorani Kurdish, Macedonian, Ta'izzi-Adeni Arabic, Bulgarian, Greek, Tunisian Arabic, Turkish, Maltese, Najdi Arabic, Arabic, Hebrew, Mesopotamian Arabic, North Levantine Arabic |
| Esperanto | Venetian, Finnish, Catalan, Danish, Ewe, Greek, Ligurian, Occitan, Bulgarian, Welsh, Bengali, Dutch, South Azerbaijani, Luxembourgish, Egyptian Arabic, Irish, Amharic, Basque, German, French |
| Estonian | Ewe, Basque, Czech, Greek, Danish, Polish, Norwegian Nynorsk, Bulgarian, Bengali, Belarusian, South Azerbaijani, Norwegian, Egyptian Arabic, Lithuanian, Amharic, Latvian, Swedish, German, Hungarian, Finnish |
| Ewe | Umbundu, Kamba (Kenya), Wolof, Luganda, Kinyarwanda, Bambara, Hausa, Tamasheq, Luba-Lulua, Kikuyu, Dyula, Chichewa, Zulu, Sango, Lingala, Mossi, Kabiyè, Yoruba, Igbo, Fon |
| Faroese | Greek, Estonian, Bulgarian, Bengali, Esperanto, Yiddish, Tok Pisin, French, Welsh, Afrikaans, Norwegian Nynorsk, German, Scottish Gaelic, Luxembourgish, Irish, Dutch, Swedish, Danish, Norwegian, Icelandic |
| Fijian | Cantonese, Korean, Japanese, Minangkabau, Malagasy, Malay, Achinese, Tok Pisin, Pangasinan, Banjar, Indonesian, Waray (Philippines), Sundanese, Javanese, Balinese, Buginese, Ilocano, Cebuano, Samoan, Maori |

| Target language | Auxiliary languages |
| --- | --- |
| Filipino[†] | Magahi, Sepedi, Luba-Lulua, Czech, Khmer, Tswana, Tamazight, Lithuanian, Lingala, Aymara, Swahili, Tajik, Chichewa, Venetian, Swedish, Ewe, North Levantine Arabic, Finnish, Fon, Mandarin Chinese |
| Finnish | Tatar, Basque, Faroese, Greek, Polish, Danish, Belarusian, Bulgarian, Bengali, Lithuanian, South Azerbaijani, Norwegian, Egyptian Arabic, Latvian, German, Norwegian Nynorsk, Amharic, Swedish, Hungarian, Estonian |
| Fon | Umbundu, Kamba (Kenya), Wolof, Luganda, Kinyarwanda, Bambara, Tamasheq, Hausa, Luba-Lulua, Kikuyu, Dyula, Chichewa, Zulu, Sango, Lingala, Mossi, Kabiyè, Igbo, Yoruba, Ewe |
| French | Romanian, Sicilian, Irish, Papiamento, Italian, Welsh, Basque, German, Dutch, Galician, Luxembourgish, Haitian Creole, Esperanto, Asturian, Spanish, Venetian, Portuguese, Catalan, Occitan, Ligurian |
| Friulian[†] | Spanish, Chichewa, Italian, Chhattisgarhi, Mossi, Uyghur, Macedonian, Slovak, Odia (Oriya), French, Haitian Creole, Sorani Kurdish, Tok Pisin, Indonesian, Latgalian, Nepali, Icelandic, Samoan, Ayacucho Quechua, Dari |
| Fulfulde[†] | Santali, Catalan, Ta'izzi-Adeni Arabic, Esperanto, Basque, Mandarin Chinese, Arabic in Latin script, Balinese, Myanmar (Burmese), Kachin, Xhosa, Albanian, Meiteilon (Manipuri), Italian, Dari, Dzongkha, Norwegian, Pangasinan, Assamese, Swati |
| Galician | Luxembourgish, Romanian, German, Sicilian, Kabyle, Haitian Creole, Esperanto, Welsh, Irish, Italian, Venetian, Basque, Papiamento, Ligurian, Occitan, French, Catalan, Spanish, Asturian, Portuguese |
| Georgian | Finnish, Arabic, Turkmen, Ewe, Turkish, Basque, Najdi Arabic, North Levantine Arabic, German, Mesopotamian Arabic, Hebrew, Sorani Kurdish, Bengali, Greek, Kurdish (Kurmanji), Amharic, Bulgarian, Armenian, Egyptian Arabic, South Azerbaijani |
| German | Italian, French, Swedish, Ligurian, Hungarian, Norwegian, Faroese, Polish, Croatian, Icelandic, Yiddish, Tok Pisin, Slovak, Venetian, Danish, Slovenian, Afrikaans, Czech, Dutch, Luxembourgish |
| Greek | Punjabi, Turkish, Lithuanian, Croatian, Kashmiri, Hungarian, Icelandic, Ukrainian, Armenian, Bosnian, Hindi, Irish, French, Albanian, Serbian, Macedonian, German, Bengali, Romanian, Bulgarian |
| Guarani[†] | Malayalam, Lingala, Ukrainian, Aymara, Galician, Luba-Lulua, Zulu, Bashkir, Sepedi, Chhattisgarhi, Arabic, Tok Pisin, Thai, Tigrinya, Japanese, Arabic in Latin script, Mizo, Najdi Arabic, Malay, Egyptian Arabic |
| Gujarati | Malayalam, Assamese, Magahi, Kannada, Sinhala, Telugu, Bhojpuri, Odia (Oriya), Maithili, Bengali, Nepali, Sanskrit, Marathi, Kashmiri, Sindhi, Chhattisgarhi, Punjabi, Awadhi, Urdu, Hindi |
| Haitian Creole | Sicilian, Scottish Gaelic, Italian, Bambara, Occitan, Icelandic, Catalan, Wolof, Irish, Aymara, Ayacucho Quechua, Ligurian, Venetian, Yiddish, Spanish, French, Asturian, Portuguese, Galician, Papiamento |
| Hausa | Bambara, Lingala, Arabic, Dyula, Mesopotamian Arabic, Sango, North Levantine Arabic, Mossi, Somali, Ewe, Kabiyè, Fon, Hebrew, Igbo, Egyptian Arabic, Maltese, Amharic, Yoruba, Tunisian Arabic, Tamasheq |
| Hebrew | Somali, Hausa, Georgian, Greek, Amharic, Armenian, Bulgarian, Kurdish (Kurmanji), Ta'izzi-Adeni Arabic, Sorani Kurdish, Turkish, South Azerbaijani, Tigrinya, Tunisian Arabic, Maltese, Najdi Arabic, Arabic, Mesopotamian Arabic, North Levantine Arabic, Egyptian Arabic |
| Hindi | Kannada, Santali, Telugu, Assamese, Sinhala, Magahi, Odia (Oriya), Sindhi, Bhojpuri, Maithili, Bengali, Kashmiri, Marathi, Nepali, Sanskrit, Urdu, Chhattisgarhi, Punjabi, Gujarati, Awadhi |
| Hungarian | Basque, Polish, Venetian, Czech, Bosnian, Ukrainian, Bengali, Slovenian, South Azerbaijani, Egyptian Arabic, Romanian, Amharic, Greek, Serbian, Croatian, Estonian, Finnish, Slovak, German, Bulgarian |
| Icelandic | Greek, Bulgarian, Finnish, Bengali, Esperanto, Yiddish, Tok Pisin, Afrikaans, Welsh, French, Norwegian Nynorsk, Luxembourgish, German, Scottish Gaelic, Irish, Dutch, Swedish, Danish, Norwegian, Faroese |
| Igbo | Tamasheq, Sepedi, Dyula, Umbundu, Sango, Rundi, Kamba (Kenya), Mossi, Kabiyè, Hausa, Kikuyu, Luganda, Ewe, Fon, Kinyarwanda, Chichewa, Zulu, Luba-Lulua, Lingala, Yoruba |
| Ilocano | Thai, Samoan, Malagasy, Balinese, Khmer, Lao, Fijian, Malay, Achinese, Indonesian, Vietnamese, Cantonese, Sundanese, Maori, Banjar, Buginese, Waray (Philippines), Cebuano, Javanese, Pangasinan |
| Indonesian | Vietnamese, Samoan, Lao, Thai, Malagasy, Khmer, Fijian, Pangasinan, Maori, Waray (Philippines), Ilocano, Cebuano, Buginese, Achinese, Javanese, Minangkabau, Malay, Banjar, Balinese, Sundanese |
| Irish | Galician, Kashmiri, Asturian, Basque, Armenian, Hindi, Danish, Luxembourgish, Greek, Faroese, Portuguese, Dutch, Bulgarian, Esperanto, Icelandic, Bengali, German, French, Welsh, Scottish Gaelic |
| Italian | Papiamento, Serbian, Hungarian, Romanian, Asturian, Haitian Creole, Albanian, Galician, Croatian, Spanish, German, Bosnian, Portuguese, Slovenian, French, Catalan, Occitan, Ligurian, Venetian, Sicilian |
| Japanese | Finnish, Kachin, Ewe, Lao, Vietnamese, Basque, Greek, Cebuano, Waray (Philippines), German, Pangasinan, Bulgarian, Bengali, Ilocano, Cantonese, South Azerbaijani, Mandarin Chinese, Egyptian Arabic, Amharic, Korean |
| Javanese | Vietnamese, Thai, Lao, Samoan, Khmer, Malagasy, Pangasinan, Waray (Philippines), Fijian, Maori, Minangkabau, Cebuano, Ilocano, Malay, Banjar, Balinese, Buginese, Achinese, Indonesian, Sundanese |
| Kabiyè | Umbundu, Kamba (Kenya), Luganda, Kinyarwanda, Wolof, Bambara, Kikuyu, Hausa, Luba-Lulua, Tamasheq, Chichewa, Dyula, Zulu, Lingala, Sango, Igbo, Yoruba, Fon, Ewe, Mossi |
| Kabuverdianu[†] | Albanian, Achinese in Arabic script, Venetian, Malagasy, Najdi Arabic, Fulfulde, Marathi, Tamil, Xhosa, Sicilian, Slovak, Bashkir, Italian, Irish, Georgian, Samoan, Achinese, Fijian, Magahi, Tigrinya |

| Target language | Auxiliary languages |
| --- | --- |
| Kabyle | Asturian, Arabic, Italian, Portuguese, Mesopotamian Arabic, North Levantine Arabic, Somali, Basque, Ligurian, Occitan, Hebrew, Hausa, Sicilian, Spanish, Egyptian Arabic, Amharic, Catalan, Tamasheq, Maltese, Tunisian Arabic |
| Kachin | Nepali, German, Vietnamese, Magahi, Bulgarian, Odia (Oriya), Maithili, South Azerbaijani, Santali, Egyptian Arabic, Amharic, Mandarin Chinese, Assamese, Shan, Cantonese, Bengali, Tibetan, Mizo, Myanmar (Burmese), Meiteilon (Manipuri) |
| Kamba (Kenya) | Chokwe, Tsonga, Tswana, Swati, Lingala, Amharic, Sesotho, Sepedi, Nuer, Somali, Luba-Lulua, Zulu, Luo, Shona, Bemba (Zambia), Chichewa, Rundi, Kinyarwanda, Luganda, Kikuyu |
| Kannada | Ewe, Sindhi, Basque, Greek, Odia (Oriya), German, Gujarati, Bulgarian, Chhattisgarhi, South Azerbaijani, Hindi, Sinhala, Bengali, Sanskrit, Egyptian Arabic, Amharic, Marathi, Tamil, Telugu, Malayalam |
| Kanuri[†] | Tsonga, Tunisian Arabic, Norwegian Nynorsk, Khmer, Dutch, Urdu, Macedonian, Lingala, Ewe, Fijian, Dinka, Odia (Oriya), Faroese, Marathi, Belarusian, Wolof, Tigrinya, Banjar in Arabic script, Mesopotamian Arabic, Estonian |
| Kanuri in Arabic script[†] | Urdu, Uzbek, Persian, Odia (Oriya), Tsonga, Kashmiri, Irish, Achinese, Maori, Dari, North Levantine Arabic, Slovak, Lingala, Kikuyu, Banjar in Arabic script, Banjar, Mandarin Chinese, Telugu, Kyrghyz, Ilocano |
| Kashmiri | Marathi, Magahi, Sanskrit, Uyghur, Assamese, Kazakh, Odia (Oriya), Bhojpuri, Sinhala, Maithili, Urdu, Kyrghyz, Bengali, Gujarati, Tajik, Awadhi, Nepali, Hindi, Sindhi, Punjabi |
| Kashmiri in Devanagari script | Marathi, Magahi, Sanskrit, Uyghur, Assamese, Kazakh, Odia (Oriya), Bhojpuri, Sinhala, Maithili, Urdu, Kyrghyz, Bengali, Gujarati, Tajik, Awadhi, Nepali, Hindi, Sindhi, Punjabi |
| Kazakh | Kurdish (Kurmanji), Sindhi, German, Armenian, Bulgarian, Georgian, Bengali, Egyptian Arabic, Amharic, Punjabi, Russian, Turkish, Kashmiri, Tajik, Tatar, South Azerbaijani, Uyghur, Turkmen, Bashkir, Kyrghyz |
| Khmer | Kachin, Javanese, Basque, Myanmar (Burmese), Greek, German, Malay, Cantonese, Bulgarian, Minangkabau, Shan, South Azerbaijani, Achinese, Egyptian Arabic, Amharic, Bengali, Thai, Santali, Lao, Vietnamese |
| Kikuyu | Chokwe, Tsonga, Tswana, Swati, Sesotho, Amharic, Lingala, Somali, Sepedi, Luba-Lulua, Nuer, Zulu, Shona, Luo, Bemba (Zambia), Chichewa, Rundi, Kinyarwanda, Luganda, Kamba (Kenya) |
| Kimbundu[†] | Irish, Chhattisgarhi, Swahili, Nepali, Kongo, Pashto, Tunisian Arabic, Norwegian Nynorsk, Uzbek, Xhosa, Bemba (Zambia), Tswana, Kashmiri in Devanagari script, South Azerbaijani, Kazakh, Azerbaijani, Kinyarwanda, Javanese, Morrocan Arabic, Latvian |
| Kinyarwanda | Umbundu, Tsonga, Tswana, Sango, Swati, Sesotho, Nuer, Sepedi, Chokwe, Zulu, Luo, Lingala, Luba-Lulua, Shona, Bemba (Zambia), Chichewa, Kamba (Kenya), Kikuyu, Luganda, Rundi |
| Kongo[†] | Bosnian, Serbian, Kashmiri, Kyrghyz, Arabic, Waray (Philippines), Amharic, Dutch, Tamazight, Marathi, Luba-Lulua, Umbundu, Mesopotamian Arabic, Samoan, Najdi Arabic, Achinese, Zulu, Tsonga, Indonesian, Balinese |
| Korean | Finnish, Lao, Ewe, Cebuano, Basque, Kachin, Greek, Vietnamese, Waray (Philippines), German, Pangasinan, Bulgarian, Ilocano, Cantonese, South Azerbaijani, Bengali, Mandarin Chinese, Egyptian Arabic, Amharic, Japanese |
| Kurdish (Kurmanji) | Awadhi, Turkmen, Assamese, Hebrew, Odia (Oriya), Nepali, North Levantine Arabic, Arabic, Sinhala, Najdi Arabic, Punjabi, Kashmiri, Armenian, Georgian, Hindi, Bengali, Mesopotamian Arabic, South Azerbaijani, Tajik, Sorani Kurdish |
| Kyrghyz | German, Hindi, Nepali, Bulgarian, Sindhi, Bengali, Egyptian Arabic, Amharic, Awadhi, Punjabi, Russian, Turkish, South Azerbaijani, Kashmiri, Tajik, Tatar, Turkmen, Bashkir, Uyghur, Kazakh |
| Lao | Achinese, Ewe, Basque, Minangkabau, Meiteilon (Manipuri), Greek, Mizo, German, Kachin, Bulgarian, Myanmar (Burmese), Cantonese, South Azerbaijani, Egyptian Arabic, Amharic, Khmer, Vietnamese, Bengali, Shan, Thai |
| Latgalian[†] | Indonesian, Kinyarwanda, Nuer, Telugu, Finnish, Polish, Balinese, Arabic in Latin script, Turkish, Sesotho, Cebuano, Tsonga, Kamba (Kenya), Awadhi, Magahi, Hungarian, Achinese, Tunisian Arabic, Malayalam, Occitan |
| Latvian | Macedonian, Norwegian Nynorsk, Bosnian, German, Danish, Norwegian, Finnish, Serbian, Swedish, Slovak, Russian, Slovenian, Croatian, Estonian, Bulgarian, Ukrainian, Czech, Belarusian, Polish, Lithuanian |
| Ligurian | Romanian, Bosnian, Basque, Croatian, Papiamento, Esperanto, Asturian, Luxembourgish, Galician, Sicilian, Slovenian, Portuguese, Haitian Creole, German, Spanish, Italian, Catalan, Occitan, French, Venetian |
| Limburgan[†] | South Azerbaijani, Morrocan Arabic, Albanian, Tok Pisin, Sinhala, Assamese, Sundanese, Khmer, Ilocano, Georgian, Somali, Sorani Kurdish, Tatar, Kabuverdianu, Irish, Romanian, Turkish, Latgalian, Kongo, Telugu |
| Lingala | Sesotho, Yoruba, Luo, Shona, Sepedi, Hausa, Bemba (Zambia), Nuer, Igbo, Zulu, Chichewa, Kamba (Kenya), Sango, Kikuyu, Rundi, Luganda, Umbundu, Kinyarwanda, Chokwe, Luba-Lulua |
| Lithuanian | Macedonian, Norwegian, Romanian, Bosnian, Hungarian, Danish, German, Swedish, Serbian, Russian, Estonian, Slovenian, Croatian, Bulgarian, Slovak, Ukrainian, Belarusian, Czech, Polish, Latvian |
| Lombard[†] | Sanskrit, Tajik, Bashkir, Myanmar (Burmese), Armenian, Spanish, Sepedi, Kyrghyz, Uyghur, Xhosa, Dzongkha, Lithuanian, Kamba (Kenya), Urdu, Ilocano, Haitian Creole, Maithili, Bhojpuri, Indonesian, Dutch |
| Luba-Lulua | Swati, Nuer, Tsonga, Sesotho, Tswana, Luo, Sepedi, Sango, Zulu, Shona, Bemba (Zambia), Kamba (Kenya), Kikuyu, Rundi, Chichewa, Luganda, Kinyarwanda, Umbundu, Chokwe, Lingala |

| Target language | Auxiliary languages |
| --- | --- |
| Luganda | Tsonga, Tswana, Sango, Amharic, Swati, Chokwe, Sesotho, Sepedi, Nuer, Zulu, Lingala, Shona, Luo, Luba-Lulua, Bemba (Zambia), Chichewa, Kamba (Kenya), Kikuyu, Kinyarwanda, Rundi |
| Luo | Chichewa, Finnish, Ewe, Luba-Lulua, Basque, Somali, Greek, Bemba (Zambia), German, Bulgarian, Bengali, Kinyarwanda, Kamba (Kenya), South Azerbaijani, Rundi, Egyptian Arabic, Luganda, Kikuyu, Amharic, Nuer |
| Luxembourgish | Bulgarian, Welsh, Bengali, Slovenian, Venetian, Swedish, Czech, Norwegian, Faroese, Ligurian, Icelandic, Occitan, Yiddish, Tok Pisin, Afrikaans, Esperanto, French, Danish, Dutch, German |
| Macedonian | Latvian, German, Sicilian, Lithuanian, Italian, Belarusian, Hungarian, Romanian, Polish, Russian, Czech, Slovak, Albanian, Ukrainian, Greek, Slovenian, Bosnian, Croatian, Serbian, Bulgarian |
| Magahi | Myanmar (Burmese), Meiteilon (Manipuri), Gujarati, Mizo, Marathi, Sanskrit, Tibetan, Sinhala, Chhattisgarhi, Kashmiri, Santali, Punjabi, Assamese, Hindi, Awadhi, Odia (Oriya), Nepali, Bengali, Bhojpuri, Maithili |
| Maithili | Myanmar (Burmese), Marathi, Gujarati, Mizo, Meiteilon (Manipuri), Sanskrit, Chhattisgarhi, Sinhala, Tibetan, Kashmiri, Santali, Punjabi, Odia (Oriya), Assamese, Hindi, Awadhi, Nepali, Bengali, Bhojpuri, Magahi |
| Malagasy | Sesotho, Buginese, Balinese, Kamba (Kenya), Bemba (Zambia), Samoan, Indonesian, Sepedi, Shona, Ilocano, Afrikaans, Fijian, Swati, Achinese, Zulu, Sundanese, Tsonga, Maori, Chichewa, Javanese |
| Malay | Vietnamese, Thai, Lao, Samoan, Malagasy, Fijian, Maori, Khmer, Pangasinan, Waray (Philippines), Ilocano, Cebuano, Minangkabau, Achinese, Buginese, Sundanese, Balinese, Javanese, Indonesian, Banjar |
| Malayalam | Ewe, Basque, Magahi, Greek, Odia (Oriya), German, Gujarati, Bulgarian, Chhattisgarhi, Sanskrit, South Azerbaijani, Hindi, Marathi, Egyptian Arabic, Amharic, Bengali, Sinhala, Telugu, Kannada, Tamil |
| Maltese | Occitan, Tigrinya, Ligurian, Amharic, Venetian, Croatian, Hebrew, Macedonian, Najdi Arabic, Ta'izzi-Adeni Arabic, Bosnian, Arabic, Italian, Mesopotamian Arabic, North Levantine Arabic, Albanian, Egyptian Arabic, Sicilian, Kabyle, Tunisian Arabic |
| Mandarin Chinese | Pangasinan, Greek, Japanese, German, Bulgarian, South Azerbaijani, Lao, Egyptian Arabic, Assamese, Amharic, Shan, Vietnamese, Korean, Bengali, Mizo, Myanmar (Burmese), Tibetan, Meiteilon (Manipuri), Kachin, Cantonese |
| Maori | Amharic, Khmer, Japanese, Minangkabau, Malagasy, Malay, Pangasinan, Tok Pisin, Waray (Philippines), Banjar, Achinese, Ilocano, Cebuano, Javanese, Buginese, Indonesian, Sundanese, Balinese, Samoan, Fijian |
| Marathi | Bhojpuri, Urdu, Assamese, Maithili, Tamil, Magahi, Nepali, Kannada, Kashmiri, Sindhi, Telugu, Awadhi, Chhattisgarhi, Bengali, Punjabi, Sinhala, Odia (Oriya), Gujarati, Sanskrit, Hindi |
| Meiteilon (Manipuri) | Nepali, Bhojpuri, German, Magahi, Bulgarian, Maithili, Odia (Oriya), South Azerbaijani, Shan, Egyptian Arabic, Amharic, Santali, Mandarin Chinese, Cantonese, Assamese, Bengali, Tibetan, Kachin, Myanmar (Burmese), Mizo |
| Mesopotamian Arabic | Turkmen, Greek, Hausa, Turkish, Bulgarian, Amharic, Armenian, Georgian, Ta'izzi-Adeni Arabic, Tigrinya, Tunisian Arabic, Kurdish (Kurmanji), Maltese, Sorani Kurdish, South Azerbaijani, Hebrew, Egyptian Arabic, Arabic, Najdi Arabic, North Levantine Arabic |
| Minangkabau | Buginese, Samoan, Vietnamese, Malagasy, Shan, Ilocano, Fijian, Maori, Myanmar (Burmese), Sinhala, Balinese, Lao, Khmer, Thai, Banjar, Indonesian, Malay, Javanese, Sundanese, Achinese |
| Minangkabau in Arabic script | Buginese, Samoan, Vietnamese, Malagasy, Shan, Ilocano, Fijian, Maori, Myanmar (Burmese), Sinhala, Balinese, Lao, Khmer, Thai, Banjar, Indonesian, Malay, Javanese, Sundanese, Achinese |
| Mizo | Nepali, Bhojpuri, German, Magahi, Bulgarian, Maithili, South Azerbaijani, Odia (Oriya), Egyptian Arabic, Amharic, Shan, Mandarin Chinese, Santali, Assamese, Cantonese, Tibetan, Bengali, Kachin, Myanmar (Burmese), Meiteilon (Manipuri) |
| Mongolian[†] | Chhattisgarhi, Welsh, Kachin, Norwegian, Marathi, Punjabi, Catalan, Kabiyè, Magahi, Tibetan, Umbundu, Faroese, Cantonese, Armenian, Russian, Dzongkha, Georgian, Turkmen, Egyptian Arabic, Shan |
| Morrocan Arabic | Bulgarian, Turkish, Tamasheq, Somali, Hausa, Armenian, Georgian, Tunisian Arabic, Kurdish (Kurmanji), Amharic, Sorani Kurdish, Maltese, South Azerbaijani, Tigrinya, Ta'izzi-Adeni Arabic, Hebrew, Egyptian Arabic, North Levantine Arabic, Najdi Arabic, Mesopotamian Arabic |
| Mossi | Kinyarwanda, Tunisian Arabic, Kamba (Kenya), Luganda, Wolof, Luba-Lulua, Kikuyu, Hausa, Bambara, Chichewa, Zulu, Tamasheq, Dyula, Lingala, Igbo, Yoruba, Sango, Fon, Ewe, Kabiyè |
| Myanmar (Burmese) | Greek, Thai, German, Magahi, Bulgarian, Maithili, South Azerbaijani, Santali, Egyptian Arabic, Amharic, Odia (Oriya), Mandarin Chinese, Assamese, Shan, Cantonese, Bengali, Tibetan, Kachin, Meiteilon (Manipuri), Mizo |
| Najdi Arabic | Tamasheq, Bulgarian, Somali, Turkish, Hausa, Armenian, Georgian, Kurdish (Kurmanji), Tunisian Arabic, Amharic, Sorani Kurdish, Maltese, South Azerbaijani, Tigrinya, Ta'izzi-Adeni Arabic, Hebrew, Egyptian Arabic, North Levantine Arabic, Arabic, Mesopotamian Arabic |
| Nepali | Mizo, Sindhi, Meiteilon (Manipuri), Gujarati, Marathi, Chhattisgarhi, Sanskrit, Tibetan, Santali, Sinhala, Magahi, Kashmiri, Punjabi, Bhojpuri, Odia (Oriya), Assamese, Maithili, Hindi, Awadhi, Bengali |
| North Levantine Arabic | Somali, Hausa, Georgian, Greek, Tigrinya, Amharic, Armenian, Bulgarian, Kurdish (Kurmanji), Ta'izzi-Adeni Arabic, Sorani Kurdish, South Azerbaijani, Turkish, Tunisian Arabic, Maltese, Arabic, Najdi Arabic, Hebrew, Mesopotamian Arabic, Egyptian Arabic |
| Norwegian | Irish, Bulgarian, Polish, Bengali, Scottish Gaelic, Yiddish, Finnish, Tok Pisin, Lithuanian, Afrikaans, Latvian, Estonian, Luxembourgish, German, Norwegian Nynorsk, Icelandic, Dutch, Faroese, Danish, Swedish |

| Target language | Auxiliary languages |
| --- | --- |
| Norwegian Nynorsk | Dutch, Ewe, Lithuanian, Basque, Irish, Scottish Gaelic, Greek, Latvian, Faroese, Danish, Bulgarian, German, Bengali, South Azerbaijani, Estonian, Egyptian Arabic, Swedish, Amharic, Norwegian, Finnish |
| Nuer | Finnish, Ta'izzi-Adeni Arabic, Somali, Ewe, Basque, Sango, Greek, Kamba (Kenya), German, Kinyarwanda, Rundi, Bulgarian, Bengali, Tigrinya, South Azerbaijani, Kikuyu, Luganda, Egyptian Arabic, Amharic, Luo |
| Occitan | Kabyle, Romanian, Croatian, Slovenian, Sicilian, Basque, Haitian Creole, Esperanto, Luxembourgish, Papiamento, Asturian, German, Galician, Portuguese, Italian, Venetian, Spanish, French, Ligurian, Catalan |
| Odia (Oriya) | Tibetan, Myanmar (Burmese), Meiteilon (Manipuri), Gujarati, Marathi, Mizo, Sanskrit, Sinhala, Punjabi, Kashmiri, Santali, Chhattisgarhi, Bhojpuri, Awadhi, Hindi, Magahi, Nepali, Maithili, Assamese, Bengali |
| Oromo[†] | Filipino, Shan, Tunisian Arabic, Tibetan, Mongolian, South Levantine Arabic, Crimean Tatar in Latin script, Kongo, Luba-Lulua, Silesian, Lingala, Ligurian, Kinyarwanda, Meiteilon (Manipuri), Latvian, Lao, Turkmen, Egyptian Arabic, Maori, Maithili |
| Pangasinan | Thai, Samoan, Malagasy, Balinese, Khmer, Indonesian, Lao, Fijian, Achinese, Vietnamese, Malay, Cantonese, Sundanese, Maori, Banjar, Buginese, Waray (Philippines), Cebuano, Javanese, Ilocano |
| Papiamento | Dyula, Sicilian, Italian, Ligurian, Venetian, Icelandic, Irish, Bambara, Occitan, French, Wolof, Catalan, Aymara, Yiddish, Ayacucho Quechua, Spanish, Asturian, Portuguese, Haitian Creole, Galician |
| Pashto[†] | Kongo, Malagasy, Kabiyè, Galician, Belarusian, Sinhala, Mossi, Korean, Sorani Kurdish, Friulian, Tatar, Tunisian Arabic, North Levantine Arabic, Japanese, Luba-Lulua, Malay, Xhosa, Swati, Sanskrit, Mandarin Chinese |
| Persian[†] | Luxembourgish, Wolof, Ukrainian, Bengali, Sesotho, Spanish, Tamasheq in Tifinagh script, Scottish Gaelic, Tamazight, Telugu, Marathi, Luba-Lulua, Sundanese, Buginese, Italian, Ligurian, Kashmiri in Devanagari script, Nuer, Chichewa, Silesian |
| Polish | Swedish, Bengali, Venetian, Macedonian, Romanian, Danish, Bosnian, Hungarian, Russian, Bulgarian, Latvian, German, Serbian, Croatian, Lithuanian, Slovenian, Ukrainian, Belarusian, Slovak, Czech |
| Portuguese | Romanian, Luxembourgish, German, Welsh, Sicilian, Irish, Kabyle, Haitian Creole, Esperanto, Italian, Venetian, Papiamento, Basque, Ligurian, Occitan, French, Catalan, Spanish, Asturian, Galician |
| Punjabi | Kyrghyz, Santali, Tajik, Chhattisgarhi, Assamese, Marathi, Sinhala, Odia (Oriya), Magahi, Sanskrit, Bengali, Bhojpuri, Urdu, Maithili, Gujarati, Awadhi, Nepali, Hindi, Sindhi, Kashmiri |
| Romanian | Albanian, Bosnian, Sicilian, Croatian, Occitan, Macedonian, Haitian Creole, Slovak, Galician, Hungarian, Venetian, Italian, Catalan, Spanish, Portuguese, Serbian, Ukrainian, Greek, French, Bulgarian |
| Rundi | Xhosa, Sango, Tsonga, Tswana, Swati, Sesotho, Sepedi, Nuer, Chokwe, Zulu, Lingala, Luo, Luba-Lulua, Shona, Bemba (Zambia), Chichewa, Kamba (Kenya), Kikuyu, Luganda, Kinyarwanda |
| Russian | Georgian, Bosnian, Latvian, Armenian, Serbian, Kashmiri, Slovenian, Turkmen, Polish, Tajik, Tatar, Croatian, Kazakh, Czech, Kyrghyz, Bulgarian, Uyghur, Ukrainian, Bashkir, Belarusian |
| Samoan | Cantonese, Korean, Minangkabau, Malagasy, Malay, Japanese, Achinese, Banjar, Tok Pisin, Pangasinan, Indonesian, Sundanese, Waray (Philippines), Javanese, Balinese, Buginese, Ilocano, Cebuano, Maori, Fijian |
| Sango | Sepedi, Chokwe, Luo, Kamba (Kenya), Chichewa, Zulu, Rundi, Nuer, Mossi, Hausa, Fon, Kikuyu, Luba-Lulua, Luganda, Kinyarwanda, Ewe, Yoruba, Kabiyè, Lingala, Igbo |
| Sanskrit | Tamil, Urdu, Assamese, Bhojpuri, Maithili, Kannada, Magahi, Sinhala, Kashmiri, Sindhi, Telugu, Nepali, Bengali, Chhattisgarhi, Punjabi, Odia (Oriya), Awadhi, Gujarati, Marathi, Hindi |
| Santali | Awadhi, Meiteilon (Manipuri), Basque, Greek, Mizo, German, Tibetan, Nepali, Bulgarian, Odia (Oriya), South Azerbaijani, Assamese, Bhojpuri, Egyptian Arabic, Amharic, Magahi, Vietnamese, Maithili, Khmer, Bengali |
| Sardinian[†] | Friulian, Kashmiri, Assamese, Haitian Creole, Chichewa, Armenian, Occitan, Tumbuka, Gujarati, Bemba (Zambia), Umbundu, Mizo, Mesopotamian Arabic, Tunisian Arabic, Shan, Punjabi, Maltese, Catalan, Kabiyè, Luxembourgish |
| Scottish Gaelic | Lithuanian, Swedish, Luxembourgish, Kashmiri, Norwegian Nynorsk, Armenian, Hindi, Esperanto, Greek, Norwegian, Danish, Dutch, Bulgarian, Faroese, Bengali, German, Icelandic, French, Welsh, Irish |
| Sepedi | Lingala, Malagasy, Umbundu, Luba-Lulua, Luganda, Chokwe, Rundi, Kamba (Kenya), Kikuyu, Kinyarwanda, Afrikaans, Bemba (Zambia), Shona, Chichewa, Xhosa, Tsonga, Tswana, Sesotho, Swati, Zulu |
| Serbian | Latvian, Italian, Venetian, Lithuanian, German, Belarusian, Russian, Albanian, Hungarian, Polish, Romanian, Czech, Slovak, Greek, Ukrainian, Slovenian, Croatian, Bosnian, Macedonian, Bulgarian |
| Sesotho | Lingala, Malagasy, Umbundu, Luba-Lulua, Chokwe, Luganda, Rundi, Kamba (Kenya), Afrikaans, Kikuyu, Kinyarwanda, Bemba (Zambia), Shona, Chichewa, Tsonga, Xhosa, Tswana, Zulu, Swati, Sepedi |
| Shan | Finnish, Santali, Ewe, Basque, Tibetan, Greek, Vietnamese, German, Bulgarian, South Azerbaijani, Assamese, Meiteilon (Manipuri), Egyptian Arabic, Amharic, Mizo, Kachin, Myanmar (Burmese), Bengali, Lao, Thai |
| Shona | Luo, Lingala, Umbundu, Luba-Lulua, Chokwe, Xhosa, Rundi, Luganda, Kamba (Kenya), Afrikaans, Kikuyu, Kinyarwanda, Sesotho, Swati, Tswana, Bemba (Zambia), Tsonga, Sepedi, Zulu, Chichewa |
| Sicilian | Greek, Asturian, Bulgarian, Croatian, Kabyle, Haitian Creole, Macedonian, Galician, Bosnian, Spanish, Albanian, Portuguese, French, Tunisian Arabic, Maltese, Ligurian, Occitan, Catalan, Venetian, Italian |

| Target language | Auxiliary languages |
| --- | --- |
| Silesian[†] | Chhattisgarhi, Scottish Gaelic, Morrocan Arabic, Banjar in Arabic script, Haitian Creole, Japanese, Kongo, Ilocano, Aymara, Venetian, Telugu, Guarani, Latgalian, Hungarian, Tigrinya, South Azerbaijani, Sardinian, Gujarati, Luo, Sanskrit |
| Sindhi | Turkmen, Magahi, Telugu, Bhojpuri, Assamese, Chhattisgarhi, Maithili, Tajik, Odia (Oriya), Sinhala, Bengali, Nepali, Marathi, Sanskrit, Awadhi, Urdu, Gujarati, Hindi, Kashmiri, Punjabi |
| Sinhala | Achinese, Minangkabau, Maithili, Magahi, Awadhi, Assamese, Nepali, Telugu, Punjabi, Kannada, Kashmiri, Chhattisgarhi, Malayalam, Tamil, Gujarati, Sanskrit, Marathi, Bengali, Odia (Oriya), Hindi |
| Slovak | Italian, Bengali, Greek, Latvian, Venetian, Romanian, Lithuanian, Macedonian, Russian, Hungarian, German, Belarusian, Bosnian, Serbian, Bulgarian, Ukrainian, Slovenian, Croatian, Polish, Czech |
| Slovenian | Latvian, Lithuanian, Albanian, Occitan, Belarusian, Ligurian, Russian, Italian, Ukrainian, Hungarian, Macedonian, Venetian, Bulgarian, Polish, German, Slovak, Czech, Serbian, Bosnian, Croatian |
| Somali | Bemba (Zambia), Kinyarwanda, Rundi, Tunisian Arabic, Luganda, Mesopotamian Arabic, Tamasheq, Nuer, North Levantine Arabic, Luo, Maltese, Hebrew, Kikuyu, Kamba (Kenya), Hausa, Egyptian Arabic, Arabic, Tigrinya, Ta'izzi-Adeni Arabic, Amharic |
| Sorani Kurdish | Awadhi, Turkmen, Assamese, Hebrew, Odia (Oriya), Nepali, North Levantine Arabic, Arabic, Sinhala, Najdi Arabic, Punjabi, Kashmiri, Armenian, Georgian, Hindi, Bengali, Mesopotamian Arabic, South Azerbaijani, Tajik, Kurdish (Kurmanji) |
| South Azerbaijani | North Levantine Arabic, Hebrew, Greek, Amharic, Bulgarian, Arabic, Uyghur, Najdi Arabic, Mesopotamian Arabic, Armenian, Georgian, Sorani Kurdish, Kyrghyz, Egyptian Arabic, Kurdish (Kurmanji), Tatar, Bashkir, Kazakh, Turkish, Turkmen |
| South Levantine Arabic[†] | Kamba (Kenya), Ilocano, Dutch, Bemba (Zambia), Mossi, Norwegian, Sorani Kurdish, Cebuano, Kyrghyz, Bambara, Turkish, Meiteilon (Manipuri), Kannada, Samoan, Spanish, Sesotho, Crimean Tatar in Latin script, Tsonga, Tamil, Bosnian |
| Spanish | Bengali, Romanian, German, Tunisian Arabic, Luxembourgish, Esperanto, Haitian Creole, Sicilian, Kabyle, Papiamento, Venetian, Basque, Italian, Ligurian, French, Occitan, Catalan, Galician, Asturian, Portuguese |
| Sundanese | Vietnamese, Samoan, Lao, Malagasy, Thai, Pangasinan, Waray (Philippines), Khmer, Fijian, Maori, Ilocano, Cebuano, Buginese, Minangkabau, Malay, Banjar, Javanese, Achinese, Balinese, Indonesian |
| Swahili[†] | Danish, Balinese, Thai, Irish, Yoruba, Arabic in Latin script, Russian, Yiddish, Bosnian, Tumbuka, Waray (Philippines), Arabic, Malagasy, Korean, Portuguese, Occitan, Sundanese, Indonesian, Galician, Basque |
| Swati | Lingala, Malagasy, Umbundu, Luba-Lulua, Chokwe, Luganda, Rundi, Kamba (Kenya), Kikuyu, Kinyarwanda, Afrikaans, Bemba (Zambia), Shona, Chichewa, Tswana, Sesotho, Xhosa, Tsonga, Sepedi, Zulu |
| Swedish | Bulgarian, Bengali, Czech, Polish, Yiddish, Belarusian, Tok Pisin, Finnish, Afrikaans, Luxembourgish, Latvian, Norwegian Nynorsk, Lithuanian, Icelandic, Estonian, Dutch, German, Faroese, Danish, Norwegian |
| Ta'izzi-Adeni Arabic | Rundi, South Azerbaijani, Tamasheq, Hausa, Luganda, Luo, Kamba (Kenya), Kikuyu, Tunisian Arabic, Nuer, Maltese, North Levantine Arabic, Hebrew, Mesopotamian Arabic, Egyptian Arabic, Somali, Tigrinya, Arabic, Amharic, Najdi Arabic |
| Taiwanese Mandarin in Traditional script | Pangasinan, Greek, Japanese, German, Bulgarian, South Azerbaijani, Lao, Egyptian Arabic, Assamese, Amharic, Shan, Vietnamese, Korean, Bengali, Mizo, Myanmar (Burmese), Tibetan, Meiteilon (Manipuri), Kachin, Cantonese |
| Tajik | South Azerbaijani, Assamese, Russian, Odia (Oriya), Sinhala, Uyghur, Gujarati, Turkmen, Urdu, Bengali, Sindhi, Kyrghyz, Awadhi, Nepali, Kazakh, Sorani Kurdish, Kurdish (Kurmanji), Hindi, Punjabi, Kashmiri |
| Tamasheq | Arabic, Igbo, Wolof, Mesopotamian Arabic, North Levantine Arabic, Somali, Yoruba, Ewe, Fon, Hebrew, Bambara, Egyptian Arabic, Kabiyè, Amharic, Dyula, Mossi, Maltese, Tunisian Arabic, Kabyle, Hausa |
| Tamasheq in Tifinagh script | Arabic, Igbo, Wolof, Mesopotamian Arabic, North Levantine Arabic, Somali, Yoruba, Ewe, Fon, Hebrew, Bambara, Egyptian Arabic, Kabiyè, Amharic, Dyula, Mossi, Maltese, Tunisian Arabic, Kabyle, Hausa |
| Tamazight[†] | Estonian, Somali, Afrikaans, Kabyle, Samoan, Punjabi, Indonesian, Buginese, Egyptian Arabic, Icelandic, Magahi, Belarusian, Norwegian Nynorsk, Sango, Persian, Oromo, Tumbuka, Norwegian, Umbundu, Kashmiri in Devanagari script |
| Tamil | Ewe, Basque, Magahi, Greek, Gujarati, German, Odia (Oriya), Bulgarian, Chhattisgarhi, Hindi, Sanskrit, South Azerbaijani, Marathi, Egyptian Arabic, Amharic, Sinhala, Bengali, Telugu, Kannada, Malayalam |
| Tatar | Ukrainian, Bengali, Bulgarian, Georgian, Egyptian Arabic, Amharic, Armenian, Uyghur, Estonian, Lithuanian, Latvian, Belarusian, Finnish, Turkmen, Kyrghyz, Russian, Turkish, South Azerbaijani, Kazakh, Bashkir |
| Telugu | Ewe, Basque, Magahi, Greek, Gujarati, Odia (Oriya), German, Sinhala, Bulgarian, South Azerbaijani, Egyptian Arabic, Chhattisgarhi, Amharic, Hindi, Sanskrit, Bengali, Marathi, Malayalam, Tamil, Kannada |
| Thai | Cantonese, Ewe, Meiteilon (Manipuri), Basque, Greek, Kachin, Mizo, German, Achinese, Bulgarian, Minangkabau, South Azerbaijani, Myanmar (Burmese), Egyptian Arabic, Amharic, Vietnamese, Khmer, Bengali, Shan, Lao |
| Tibetan | Odia (Oriya), German, Bulgarian, Awadhi, South Azerbaijani, Egyptian Arabic, Magahi, Amharic, Bhojpuri, Mandarin Chinese, Nepali, Maithili, Santali, Cantonese, Assamese, Kachin, Myanmar (Burmese), Bengali, Mizo, Meiteilon (Manipuri) |

| Target language | Auxiliary languages |
|---|---|
| Tigrinya | South Azerbaijani, Rundi, Tamasheq, Kamba (Kenya), Hausa, Luo, Luganda, Kikuyu, Tunisian Arabic, Maltese, Nuer, North Levantine Arabic, Mesopotamian Arabic, Somali, Najdi Arabic, Egyptian Arabic, Arabic, Hebrew, Ta'izzi-Adeni Arabic, Amharic |
| Tok Pisin | Indonesian, Danish, Pangasinan, Swedish, Norwegian, Ilocano, Malay, Faroese, Icelandic, Banjar, Luxembourgish, Balinese, Yiddish, Fijian, Dutch, Waray (Philippines), Afrikaans, Buginese, German, Cebuano |
| Tsonga | Lingala, Malagasy, Umbundu, Luba-Lulua, Chokwe, Luganda, Rundi, Kamba (Kenya), Kikuyu, Kinyarwanda, Bemba (Zambia), Afrikaans, Shona, Chichewa, Sesotho, Xhosa, Tswana, Sepedi, Swati, Zulu |
| Tswana | Lingala, Malagasy, Umbundu, Kamba (Kenya), Luba-Lulua, Kikuyu, Chokwe, Rundi, Luganda, Kinyarwanda, Bemba (Zambia), Afrikaans, Shona, Chichewa, Xhosa, Tsonga, Swati, Sesotho, Zulu, Sepedi |
| Tumbuka† | Papiamento, Odia (Oriya), Irish, Achinese in Arabic script, Kachin, Faroese, Cantonese, Ligurian, Banjar in Arabic script, Kimbundu, Bengali, Meiteilon (Manipuri), Fijian, Chokwe, Nuer, Morrocan Arabic, Hebrew, Mongolian, Afrikaans, Tswana |
| Tunisian Arabic | Hausa, Bosnian, Tigrinya, Amharic, Albanian, Hebrew, Spanish, Najdi Arabic, Occitan, Ta'izzi-Adeni Arabic, Ligurian, Italian, Arabic, Mesopotamian Arabic, Catalan, North Levantine Arabic, Sicilian, Egyptian Arabic, Kabyle, Maltese |
| Turkish | Albanian, Georgian, Bengali, Armenian, Amharic, Serbian, Romanian, Uyghur, Turkmen, Tatar, Kazakh, Macedonian, Hebrew, Bashkir, Kyrghyz, North Levantine Arabic, South Azerbaijani, Greek, Egyptian Arabic, Bulgarian |
| Turkmen | German, Urdu, Bulgarian, Bengali, Mesopotamian Arabic, Egyptian Arabic, Kashmiri, Amharic, Armenian, Sorani Kurdish, Georgian, Tatar, Kurdish (Kurmanji), Turkish, Uyghur, Tajik, Bashkir, Kyrghyz, Kazakh, South Azerbaijani |
| Ukrainian | French, Albanian, Bengali, Latvian, German, Lithuanian, Greek, Hungarian, Bosnian, Macedonian, Romanian, Russian, Slovenian, Croatian, Czech, Slovak, Polish, Serbian, Belarusian, Bulgarian |
| Umbundu | Sango, Swati, Sesotho, Kamba (Kenya), Igbo, Tsonga, Afrikaans, Kikuyu, Luganda, Rundi, Bemba (Zambia), Sepedi, Tswana, Kinyarwanda, Zulu, Shona, Chichewa, Lingala, Luba-Lulua, Chokwe |
| Urdu | Magahi, Assamese, Odia (Oriya), Telugu, Bhojpuri, Maithili, Turkmen, Sinhala, Tajik, Bengali, Nepali, Marathi, Sanskrit, Chhattisgarhi, Sindhi, Awadhi, Kashmiri, Punjabi, Gujarati, Hindi |
| Uyghur | German, Bhojpuri, Bulgarian, Tibetan, Egyptian Arabic, Amharic, Awadhi, Bengali, Nepali, Tatar, Punjabi, Russian, Turkish, Kashmiri, Tajik, South Azerbaijani, Bashkir, Turkmen, Kazakh, Kyrghyz |
| Uzbek† | Bhojpuri, Hebrew, Fijian, Romanian, French, Tumbuka, Spanish, Irish, Banjar in Arabic script, Sundanese, Swati, Thai, Lao, Maori, Bulgarian, Finnish, Tamasheq in Tifinagh script, Slovak, Ayacucho Quechua, Danish |
| Venetian | Slovak, Papiamento, Hungarian, Romanian, Sicilian, Asturian, Czech, Galician, Bosnian, Spanish, Portuguese, Croatian, Haitian Creole, Slovenian, French, Catalan, German, Occitan, Italian, Ligurian |
| Vietnamese | Ewe, Meiteilon (Manipuri), Basque, Greek, Ilocano, Pangasinan, German, Myanmar (Burmese), Bulgarian, Kachin, South Azerbaijani, Shan, Cantonese, Egyptian Arabic, Thai, Amharic, Lao, Santali, Bengali, Khmer |
| Waray (Philippines) | Minangkabau, Lao, Samoan, Khmer, Vietnamese, Malagasy, Cantonese, Fijian, Achinese, Maori, Balinese, Malay, Indonesian, Banjar, Sundanese, Pangasinan, Buginese, Javanese, Ilocano, Cebuano |
| Welsh | Lithuanian, Galician, Kashmiri, Icelandic, Armenian, Asturian, Basque, Hindi, Danish, Luxembourgish, Greek, Dutch, Bulgarian, Portuguese, Esperanto, Bengali, German, French, Scottish Gaelic, Irish |
| Wolof | Kamba (Kenya), Spanish, Galician, Sango, Kabyle, Luganda, Lingala, Hausa, Kikuyu, Chichewa, Tamasheq, Zulu, Dyula, Bambara, Mossi, Fon, Yoruba, Igbo, Kabiyè, Ewe |
| Xhosa | Lingala, Malagasy, Umbundu, Luba-Lulua, Chokwe, Luganda, Rundi, Kamba (Kenya), Afrikaans, Kikuyu, Kinyarwanda, Bemba (Zambia), Shona, Chichewa, Tswana, Tsonga, Sepedi, Sesotho, Zulu, Swati |
| Yiddish | Bengali, Portuguese, Swedish, Asturian, Galician, Welsh, Danish, Scottish Gaelic, French, Tok Pisin, Luxembourgish, Papiamento, Afrikaans, Norwegian, Haitian Creole, German, Irish, Dutch, Faroese, Icelandic |
| Yoruba | Sango, Rundi, Umbundu, Bambara, Tamasheq, Kamba (Kenya), Dyula, Hausa, Luganda, Mossi, Kinyarwanda, Kikuyu, Chichewa, Kabiyè, Luba-Lulua, Zulu, Ewe, Fon, Lingala, Igbo |
| Zulu | Lingala, Malagasy, Umbundu, Luba-Lulua, Chokwe, Luganda, Rundi, Kamba (Kenya), Kikuyu, Kinyarwanda, Afrikaans, Bemba (Zambia), Shona, Chichewa, Tswana, Sesotho, Xhosa, Tsonga, Sepedi, Swati |

Table 5: Auxiliary languages sorted from furthest to closest, based on genealogical and geographic distance documented in URIEL repository (Littell et al., 2017). Languages marked with '†' are not included in the database, for which we sample the auxiliary languages in random. Languages without script notation are in the dominant script—Achinese in Latin script, Hindi in Devanagari script, etc.

| | | FLORES-200 devtest | | | | | NTREX | | |
|---|---|---|---|---|---|---|---|---|---|
| | | BLEU↑ (n=201) | BLEU↑ (n=198) | Win% vs. teacher | Win% vs. NLLB 1.3B | Win% vs. NLLB 54B | BLEU↑ (n=112) | Win% vs. teacher | Win% vs. NLLB 1.3B |
| PaLM2 S (teacher) | | 17.4 | 17.7 | - | 58.6 | 44.2 | 20.2 | - | 75.9 |
| NLLB 1.3B distilled | | - | 16.9 | 40.8 | - | 7.0 | 18.7 | 24.1 | - |
| NLLB 54B MoE | | - | 19.4 | 55.2 | 92.9 | - | - | - | - |
| PaLM2 XXS –NTL | baseline | 11.8 | 11.9 | 35.3 | 18.2 | 12.6 | 9.2 | 10.7 | 6.2 |
| | mufu0 | 14.3 | 14.5 | 39.8 | 23.7 | 16.1 | 12.1 | 11.6 | 8.0 |
| | mufu5 | 18.7 | 18.9 | 53.2 | 63.1 | 32.7 | 17.5 | 21.4 | 25.9 |
| | mufu10 | 19.8 | 20.0 | 64.7 | 80.8 | 46.2 | 18.9 | 25.0 | 45.5 |
| | mufu20 | **20.2** | **20.5** | 66.2 | 83.8 | 52.8 | 20.1 | 31.2 | 61.6 |
| | mufu5hrl | 14.5 | 14.7 | 39.3 | 27.3 | 17.1 | 12.3 | 11.6 | 8.0 |
| | mufu5tr | 16.2 | 16.3 | 45.3 | 40.9 | 27.1 | 14.5 | 17.0 | 12.5 |
| | mufu20+5hrl | 18.8 | 19.0 | 56.2 | 67.7 | 33.7 | 18.0 | 21.4 | 28.6 |
| | distilled | 17.2 | 17.4 | 50.2 | 44.9 | 28.1 | **20.2** | 53.6 | 54.5 |
| PaLM2 XXS | baseline | 9.7 | 9.8 | 28.9 | 10.6 | 10.1 | 8.1 | 6.2 | 6.2 |
| | mufu0 | **14.9** | **15.1** | 32.8 | 27.8 | 18.1 | **15.1** | 12.5 | 15.2 |
| | mufu5 | 14.7 | 14.9 | 34.8 | 25.3 | 17.1 | 14.5 | 10.7 | 9.8 |
| | mufu10 | 13.4 | 13.6 | 34.3 | 18.2 | 14.6 | 11.9 | 8.0 | 7.1 |
| | mufu20 | 13.6 | 13.8 | 34.8 | 18.2 | 14.6 | 12.1 | 8.0 | 7.1 |
| PaLM2 XS | baseline | 2.9 | 2.9 | 8.0 | 3.5 | 2.5 | 2.7 | 1.8 | 2.7 |
| | mufu0 | 15.6 | 15.8 | 41.3 | 32.8 | 20.1 | 13.8 | 12.5 | 12.5 |
| | mufu5 | 16.1 | 16.3 | 44.3 | 36.4 | 22.1 | **14.2** | 12.5 | 13.4 |
| | mufu10 | **16.1** | **16.3** | 45.3 | 35.4 | 21.6 | 14.1 | 12.5 | 12.5 |
| | mufu20 | 16.1 | 16.3 | 45.3 | 34.8 | 21.1 | 14.1 | 11.6 | 13.4 |
| PaLM2 S | baseline | 3.9 | 3.9 | 15.4 | 4.0 | 3.0 | 3.0 | 1.8 | 2.7 |
| | mufu20 | 18.5 | 18.6 | 56.7 | 59.1 | 31.2 | 16.1 | 18.8 | 25.9 |
| | mufu20lora | **20.5** | **20.7** | 98.0 | 73.2 | 63.3 | **21.7** | 81.2 | 83.9 |
| Gemma 2B | baseline | 9.5 | 9.5 | 33.8 | 14.6 | 12.6 | 6.9 | 12.5 | 8.0 |
| | mufu0 | 16.8 | 16.9 | 40.3 | 43.4 | 25.1 | 14.5 | 16.1 | 16.1 |
| | mufu5 | 17.4 | 17.6 | 45.8 | 51.0 | 28.6 | 15.3 | 17.0 | 17.9 |
| | mufu10 | 17.6 | 17.7 | 46.3 | 56.1 | 30.2 | 15.3 | 19.6 | 19.6 |
| | mufu20 | **17.7** | **17.9** | 46.8 | 53.0 | 28.1 | **15.5** | 19.6 | 20.5 |
| Gemma 7B | baseline | 13.2 | 13.3 | 38.8 | 25.8 | 16.1 | 9.6 | 11.6 | 9.8 |
| | mufu0 | 18.6 | 18.8 | 50.7 | 59.1 | 32.7 | 15.4 | 16.1 | 22.3 |
| | mufu5 | 19.1 | 19.3 | 56.2 | 67.2 | 36.2 | 15.4 | 18.8 | 20.5 |
| | mufu10 | 19.3 | 19.4 | 58.2 | 67.7 | 37.2 | 15.4 | 17.9 | 20.5 |
| | mufu20 | **19.6** | **19.7** | 59.2 | 71.2 | 39.7 | 15.7 | 19.6 | 23.2 |
| | mufu5hrl | 18.6 | 18.8 | 51.7 | 61.6 | 33.7 | 15.2 | 17.9 | 23.2 |
| | mufu5tr | 15.3 | 15.3 | 45.3 | 34.8 | 24.1 | 10.8 | 12.5 | 7.1 |
| | mufu20+5hrl | **19.6** | **19.7** | 58.2 | 72.2 | 40.7 | 15.7 | 19.6 | 19.6 |
| | distilled | 16.6 | 16.7 | 45.8 | 36.4 | 25.1 | **18.8** | 38.4 | 52.7 |

Table 6: Mean BLEU scores, analogous to chrF scores reported in Table 2. **Bold** values are the best scores in a given model class. Red values are win rates above 50%.

## A.3 EXPERIMENTAL DETAILS

We perform full parameter updates for 25 epochs across all models, and select the final checkpoints with the best chrF scores for very-low- and low-resource languages over the validation split, which is partitioned from FLORES-200 devtest as described in Section 3.1. All Gemma models are finetuned at a learning rate of 1e-5. We set the initial learning rate to 1e-4 for PaLM2 models. When the models fail to converge, we reduce the rate to 1e-5 in the reruns. During evaluation, we greedily decode from the finetuned models and compute chrF based on the generated sequence and reference translation.

## A.4 BLEU SCORES

We report mean BLEU and overall win rates against benchmarks in Table 6, which is analogous to Table 2 in the main text. Figure 4 and Table 7 report Mufu's performance in very-low- and low-resource languages, and are analogous to Figure 2 and Table 3 respectively.

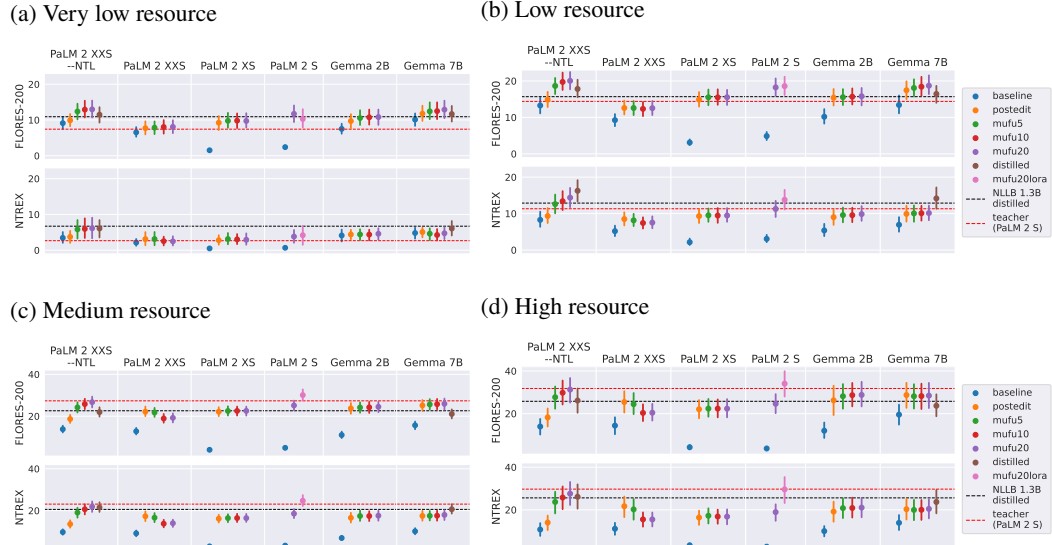

Figure 4: Mean BLEU across languages of the same resource level, analogous to Figure 2. Note that the scales of y-axes are different for the top and bottom rows. Error bars shown are 95% confidence intervals across the language pairs.

|  |  | FLORES-200 devtest | | | NTREX | |
|---|---|---|---|---|---|---|
|  |  | teacher | NLLB 1.3B | NLLB 54B | teacher | NLLB 1.3B |
| PaLM2 XXS –NTL | baseline | 61.2 | 24.8 | 14.9 | 35.5 | 6.5 |
|  | mufu0 | 66.4 | 30.1 | 18.4 | 35.5 | 9.7 |
|  | mufu5 | 81.0 | 64.6 | 39.5 | 54.8 | 25.8 |
|  | mufu10 | 92.2 | 73.5 | 52.6 | 58.1 | 35.5 |
|  | mufu20 | 92.2 | 75.2 | 50.9 | 64.5 | 41.9 |
|  | distilled | 80.2 | 54.9 | 33.3 | 83.9 | 48.4 |
| Gemma 7B | baseline | 64.7 | 35.4 | 20.2 | 32.3 | 9.7 |
|  | mufu0 | 77.6 | 50.4 | 37.7 | 35.5 | 19.4 |
|  | mufu5 | 86.2 | 60.2 | 42.1 | 35.5 | 9.7 |
|  | mufu10 | 85.3 | 59.3 | 42.1 | 35.5 | 9.7 |
|  | mufu20 | 87.9 | 64.6 | 44.7 | 38.7 | 12.9 |
|  | distilled | 76.7 | 47.8 | 30.7 | 58.1 | 45.2 |

Table 7: Win percentages by BLEU scores, analogous to Table 3, measured over the 113 low and very-low resource languages for models shown in rows against, as columns, the teacher model, NLLB 1.3B distilled and NLLB 54B MoE. Win rates above 50% are in red.

## A.5 MUFU RESULTS BY LANGUAGE PAIRS

The full results (chrF) by language pairs for PaLM2 XXS–NTL and Gemma 7B finetuned on mufu20 is reported in Table 8. The models are mostly better than the teacher and NLLB 1.3B distilled when translating into languages classified as very-low- or low-resource.

|  |  | FLORES-200 devtest | | | | NTREX | | | |
|---|---|---|---|---|---|---|---|---|---|
| target | resource | PaLM2 S (teacher) | NLLB 1.3B distilled | PaLM2 XXS pt. NTL (mufu20) | Gemma 7B (mufu20) | PaLM2 S (teacher) | NLLB 1.3B distilled | PaLM2 XXS pt. NTL (mufu20) | Gemma 7B (mufu20) |
| Achinese | VL | 31.8 | 40.7 | **47.6** | 46.7 | - | - | - | - |
| Achinese in Arabic script | VL | 5.9 | 18.0 | **27.1** | 36.6 | - | - | - | - |
| Afrikaans | M | 70.7 | 65.0 | 70.2 | 70.1 | 70.7 | 68.7 | 68.4 | 62.5 |
| Albanian | M | 62.1 | 58.4 | 60.4 | 59.0 | 59.7 | 57.8 | 57.6 | 52.3 |
| Amharic | L | 41.3 | 37.0 | 39.6 | 35.9 | 26.4 | 26.6 | 25.4 | 21.4 |

| target | resource | FLORES-200 devtest | | | | NTREX | | | |
|---|---|---|---|---|---|---|---|---|---|
| | | PaLM2 S (teacher) | NLLB 1.3B distilled | PaLM2 XXS pt. NTL (mufu20) | Gemma 7B (mufu20) | PaLM2 S (teacher) | NLLB 1.3B distilled | PaLM2 XXS pt. NTL (mufu20) | Gemma 7B (mufu20) |
| Arabic | M | 60.7 | 56.5 | 59.2 | 59.8 | 55.3 | 51.6 | 53.2 | 49.2 |
| Arabic in Latin script | VL | 27.8 | - | **33.5** | **44.1** | - | - | - | - |
| Armenian | M | 58.7 | 52.5 | 56.8 | 56.7 | 53.5 | 50.2 | 51.5 | 47.7 |
| Assamese | L | 41.6 | 37.9 | **42.5** | 40.6 | - | - | - | - |
| Asturian | VL | 61.5 | 50.5 | 60.0 | 60.2 | - | - | - | - |
| Awadhi | VL | 50.8 | 49.3 | **56.0** | **59.5** | - | - | - | - |
| Ayacucho Quechua | VL | 23.6 | 28.0 | **38.3** | 32.8 | - | - | - | - |
| Aymara | VL | 14.5 | 31.7 | **33.6** | 29.4 | - | - | - | - |
| Azerbaijani | M | 47.8 | 45.0 | 46.0 | 43.8 | 49.0 | 48.2 | 46.0 | 41.6 |
| Balinese | VL | 40.5 | 48.3 | **53.8** | **51.2** | - | - | - | - |
| Bambara | VL | 10.6 | 32.1 | 31.9 | 28.8 | - | - | - | - |
| Banjar | VL | 48.6 | 50.7 | **54.5** | **54.0** | - | - | - | - |
| Banjar in Arabic script | VL | 14.5 | 17.5 | **30.3** | **36.6** | - | - | - | - |
| Bashkir | L | 47.4 | 48.3 | **51.7** | **50.0** | 39.6 | 42.0 | **42.9** | 39.3 |
| Basque | M | 57.0 | 52.2 | 54.0 | 53.6 | 52.6 | 49.3 | 49.9 | 41.0 |
| Belarusian | M | 45.7 | 43.2 | 43.8 | 44.5 | 54.4 | 54.5 | 50.0 | 45.6 |
| Bemba (Zambia) | VL | 35.8 | 37.9 | **42.0** | **39.0** | 37.1 | 40.9 | **41.2** | 36.9 |
| Bengali | M | 52.5 | 50.7 | 51.9 | 51.6 | 52.2 | 51.5 | 50.9 | 43.3 |
| Bhojpuri | VL | 41.1 | 43.7 | **45.0** | 41.9 | - | - | - | - |
| Bosnian | M | 62.6 | 58.6 | 61.5 | 61.4 | 58.5 | 56.8 | 57.2 | 53.8 |
| Buginese | VL | 20.5 | 37.2 | **37.8** | 34.2 | - | - | - | - |
| Bulgarian | M | 68.3 | 64.1 | 66.6 | 64.6 | 59.3 | 56.9 | 57.6 | 52.7 |
| Cantonese | M | 40.1 | 18.0 | 38.3 | 31.7 | 26.2 | 18.1 | 24.9 | 22.1 |
| Catalan | M | 67.2 | 63.8 | 66.3 | 65.1 | 62.9 | 61.0 | 61.6 | 52.0 |
| Cebuano | M | 60.0 | 57.8 | **61.8** | 57.7 | - | - | - | - |
| Chhattisgarhi | VL | 50.6 | 55.8 | **57.6** | **58.8** | - | - | - | - |
| Chichewa | M | 49.2 | 48.3 | 48.7 | 44.8 | 52.2 | 51.0 | 50.5 | 44.8 |
| Chokwe | VL | 9.2 | 25.7 | 17.8 | **27.2** | - | - | - | - |
| Crimean Tatar in Latin script | VL | 38.0 | 47.3 | 39.2 | 42.1 | - | - | - | - |
| Croatian | M | 60.6 | 56.1 | 59.0 | 59.5 | 59.4 | 57.2 | 57.7 | 51.6 |
| Czech | H | 60.3 | 56.1 | 58.8 | 55.6 | 58.9 | 55.9 | 56.4 | 52.3 |
| Danish | H | 71.1 | 65.0 | 69.3 | 69.2 | 64.1 | 60.5 | 63.0 | 63.1 |
| Dari | M | 54.9 | 53.2 | 54.3 | 49.3 | 44.3 | 42.6 | 43.7 | 36.4 |
| Dinka | VL | 9.1 | 23.2 | 22.8 | **23.8** | - | - | - | - |
| Dutch | H | 59.7 | 56.3 | 58.3 | 55.1 | 63.7 | 60.7 | 61.1 | 53.3 |
| Dyula | VL | 8.0 | 18.0 | **18.4** | **21.3** | - | - | - | - |
| Dzongkha | L | 32.0 | 41.1 | **42.8** | 41.0 | 28.3 | 36.5 | **37.6** | 31.6 |
| Egyptian Arabic | VL | 49.1 | 47.9 | **51.2** | 48.3 | - | - | - | - |
| Esperanto | M | 63.4 | 62.7 | 62.8 | **63.6** | - | - | - | - |
| Estonian | M | 62.4 | 54.5 | 59.8 | 59.0 | 59.3 | 54.6 | 56.8 | 49.9 |
| Ewe | VL | 8.0 | 38.9 | 33.8 | 29.6 | 9.0 | 38.7 | 33.5 | 26.3 |
| Faroese | L | 46.0 | 45.8 | **49.6** | **48.1** | 48.7 | 50.5 | **51.9** | 44.8 |
| Fijian | L | 28.4 | 46.2 | 46.0 | 41.0 | 29.7 | 49.4 | **50.7** | 38.5 |
| Filipino | M | 64.0 | 59.9 | 63.4 | 59.1 | 64.0 | 60.9 | 61.5 | 53.0 |
| Finnish | H | 61.1 | 53.8 | 58.1 | 57.0 | 56.3 | 50.0 | 54.1 | 50.2 |
| Fon | VL | 4.2 | 20.0 | **20.1** | 18.0 | - | - | - | - |
| French | H | 73.1 | 68.9 | 72.4 | 69.7 | 64.3 | 60.4 | 62.1 | 50.7 |
| Friulian | VL | 49.2 | 57.1 | 56.5 | 54.2 | - | - | - | - |
| Fulfulde | VL | 5.7 | 23.8 | 21.8 | **24.1** | 6.0 | 27.6 | 22.0 | 22.0 |
| Galician | M | 62.5 | 60.0 | **62.6** | 61.8 | 63.7 | 62.6 | 62.6 | 59.1 |
| Georgian | M | 54.1 | 48.4 | 52.2 | 52.2 | 49.8 | 45.5 | 47.2 | 44.4 |

| target | resource | FLORES-200 devtest | | | | NTREX | | | |
|---|---|---|---|---|---|---|---|---|---|
| | | PaLM2 S (teacher) | NLLB 1.3B distilled | PaLM2 XXS pt. NTL (mufu20) | Gemma 7B (mufu20) | PaLM2 S (teacher) | NLLB 1.3B distilled | PaLM2 XXS pt. NTL (mufu20) | Gemma 7B (mufu20) |
| German | H | 67.1 | 61.8 | 66.1 | 61.5 | 62.1 | 58.5 | 60.8 | 53.2 |
| Greek | M | 54.4 | 52.3 | 53.7 | 54.3 | 59.4 | 58.1 | 57.6 | 49.7 |
| Guarani | VL | 24.3 | 39.1 | 38.8 | 34.5 | - | - | - | - |
| Gujarati | M | 53.6 | 53.5 | **54.4** | 53.4 | 48.4 | 49.3 | 48.0 | 44.6 |
| Haitian Creole | M | 54.5 | 52.7 | **56.8** | 54.4 | - | - | - | - |
| Hausa | L | 52.9 | 51.8 | 51.5 | 49.8 | 54.1 | 54.1 | 51.9 | 45.5 |
| Hebrew | M | 61.6 | 57.0 | 59.5 | 58.0 | 54.2 | 51.5 | 51.7 | 47.1 |
| Hindi | M | 59.7 | 56.0 | 58.8 | 59.5 | 52.3 | 51.3 | 51.0 | 43.2 |
| Hungarian | M | 57.8 | 53.5 | 56.2 | 55.9 | 49.7 | 46.2 | 47.7 | 42.0 |
| Icelandic | M | 52.8 | 47.9 | 50.6 | 49.7 | 54.1 | 50.2 | 52.0 | 47.3 |
| Igbo | L | 42.4 | 41.8 | 41.3 | 38.8 | 47.6 | 48.0 | 45.2 | 37.3 |
| Ilocano | L | 46.0 | 53.7 | **55.8** | 51.8 | - | - | - | - |
| Indonesian | M | 72.3 | 69.0 | 71.4 | 70.6 | 67.4 | 65.0 | 66.5 | 62.9 |
| Irish | M | 58.7 | 53.8 | 56.2 | 58.4 | 55.0 | 51.7 | 52.2 | 48.9 |
| Italian | H | 60.1 | 58.0 | 59.8 | 57.9 | 62.8 | 62.0 | 61.5 | 54.5 |
| Japanese | H | 46.6 | 30.0 | 44.0 | 38.8 | 37.9 | 27.7 | 34.9 | 28.0 |
| Javanese | L | 57.0 | 56.0 | 56.9 | 52.6 | - | - | - | - |
| Kabiyè | VL | 11.6 | 28.2 | **29.4** | 26.8 | - | - | - | - |
| Kabuverdianu | VL | 43.2 | 44.7 | **47.8** | **58.3** | - | - | - | - |
| Kabyle | VL | 15.2 | 32.1 | **32.7** | 31.4 | - | - | - | - |
| Kachin | VL | 14.0 | 37.5 | **39.9** | 35.9 | - | - | - | - |
| Kamba (Kenya) | VL | 11.2 | 28.5 | 18.6 | **30.8** | - | - | - | - |
| Kannada | M | 56.0 | 55.2 | 54.8 | 54.9 | 52.2 | 53.0 | 50.8 | 44.1 |
| Kanuri | VL | 10.6 | 25.2 | **27.2** | 24.7 | - | - | - | - |
| Kanuri in Arabic script | VL | 10.9 | 13.1 | 10.8 | **19.4** | - | - | - | - |
| Kashmiri | VL | 16.9 | 37.1 | 36.6 | 34.3 | - | - | - | - |
| Kashmiri in Devanagari script | VL | 13.6 | 18.7 | **26.6** | **29.2** | - | - | - | - |
| Kazakh | M | 58.1 | 50.1 | 56.9 | 57.1 | 48.9 | 45.2 | 48.4 | 43.7 |
| Khmer | M | 46.5 | 37.9 | 45.5 | 43.8 | 50.5 | 49.0 | 48.0 | 44.1 |
| Kikuyu | VL | 11.4 | 37.2 | 33.6 | 35.5 | - | - | - | - |
| Kimbundu | VL | 13.6 | 28.5 | **31.2** | **35.1** | - | - | - | - |
| Kinyarwanda | L | 26.3 | 48.6 | 45.2 | 38.0 | 27.9 | 47.9 | 43.4 | 33.8 |
| Kongo | VL | 21.3 | 46.9 | **48.8** | 41.0 | - | - | - | - |
| Korean | H | 40.6 | 34.4 | 37.7 | 36.5 | 37.7 | 30.2 | 33.5 | 31.1 |
| Kurdish (Kurmanji) | M | 40.5 | 39.1 | **40.7** | 38.6 | 39.2 | 39.2 | 38.3 | 34.1 |
| Kyrghyz | L | 47.6 | 44.6 | 47.5 | 45.2 | 43.6 | 43.4 | **43.6** | 39.1 |
| Lao | M | 51.4 | 49.2 | **53.7** | **52.3** | 37.0 | 38.9 | **39.6** | **46.2** |
| Latgalian | VL | 31.6 | 48.1 | **50.5** | 46.9 | - | - | - | - |
| Latvian | M | 60.4 | 50.3 | 58.0 | 57.0 | 52.8 | 45.9 | 50.5 | 49.4 |
| Ligurian | VL | 45.2 | 48.5 | **55.3** | **54.2** | - | - | - | - |
| Limburgan | VL | 49.7 | 46.8 | 48.4 | 48.4 | - | - | - | - |
| Lingala | L | 27.1 | 49.6 | **49.7** | 45.8 | - | - | - | - |
| Lithuanian | M | 60.0 | 53.2 | 57.5 | 56.5 | 55.1 | 50.6 | 52.4 | 50.5 |
| Lombard | VL | 36.3 | 36.0 | **38.9** | **40.6** | - | - | - | - |
| Luba-Lulua | VL | 15.0 | 37.5 | **38.3** | 31.9 | - | - | - | - |
| Luganda | L | 20.5 | 40.8 | 38.7 | 31.7 | - | - | - | - |
| Luo | VL | 15.9 | 40.0 | 38.5 | 34.4 | - | - | - | - |
| Luxembourgish | M | 59.1 | 55.2 | 58.5 | **59.6** | 53.4 | 52.5 | 51.2 | 49.4 |
| Macedonian | M | 65.0 | 60.3 | 63.1 | 62.5 | 62.7 | 60.2 | 60.6 | 59.8 |
| Magahi | VL | 55.2 | 58.1 | **60.5** | **63.0** | - | - | - | - |
| Maithili | L | 50.8 | 48.9 | **58.7** | **61.5** | - | - | - | - |

| target | resource | FLORES-200 devtest | | | | NTREX | | | |
|---|---|---|---|---|---|---|---|---|---|
| | | PaLM2 S (teacher) | NLLB 1.3B distilled | PaLM2 XXS pt. NTL (mufu20) | Gemma 7B (mufu20) | PaLM2 S (teacher) | NLLB 1.3B distilled | PaLM2 XXS pt. NTL (mufu20) | Gemma 7B (mufu20) |
| Malagasy | M | 57.6 | 52.4 | 55.1 | 52.7 | 52.1 | 49.5 | 49.8 | 43.4 |
| Malay | M | 70.2 | 66.7 | 69.1 | 66.8 | 66.2 | 63.6 | 65.1 | 65.6 |
| Malayalam | M | 58.1 | 50.4 | 55.8 | 55.5 | 49.6 | 44.2 | 47.7 | 45.9 |
| Maltese | M | 71.2 | 66.0 | 68.9 | 69.5 | 66.9 | 62.2 | 64.3 | 61.1 |
| Mandarin Chinese | H | 42.3 | 23.6 | 40.2 | 37.0 | 34.5 | 18.8 | 32.3 | 24.3 |
| Maori | L | 48.2 | 47.4 | **48.8** | **48.7** | 51.8 | 49.5 | 50.9 | 45.0 |
| Marathi | M | 52.2 | 47.6 | 50.7 | 52.1 | 47.7 | 45.5 | 46.2 | 45.8 |
| Meiteilon (Manipuri) | VL | 12.6 | 40.2 | 39.3 | 39.2 | - | - | - | - |
| Mesopotamian Arabic | L | 52.2 | 48.4 | **53.6** | **53.4** | - | - | - | - |
| Minangkabau | VL | 51.1 | 52.0 | **57.4** | **55.0** | - | - | - | - |
| Minangkabau in Arabic script | VL | 16.8 | - | **34.8** | **44.8** | - | - | - | - |
| Mizo | VL | 19.7 | 38.0 | **38.2** | 33.9 | - | - | - | - |
| Mongolian | M | 51.4 | 41.9 | 50.8 | 49.4 | 45.8 | 40.2 | 44.5 | 36.1 |
| Morrocan Arabic | L | 42.7 | 40.7 | **43.4** | 42.2 | - | - | - | - |
| Mossi | VL | 3.7 | 23.5 | 11.9 | 22.6 | - | - | - | - |
| Myanmar (Burmese) | M | 51.7 | 37.8 | 50.4 | 49.1 | 18.0 | 17.8 | 17.6 | 17.4 |
| Najdi Arabic | VL | 59.7 | 53.5 | 58.3 | **60.1** | - | - | - | - |
| Nepali | M | 58.4 | 50.4 | 57.2 | 56.9 | 47.4 | 44.1 | 46.0 | 42.9 |
| North Levantine Arabic | L | 52.6 | 49.3 | **57.8** | **59.9** | - | - | - | - |
| Norwegian | H | 62.5 | 59.6 | 61.6 | 60.1 | 64.3 | 61.1 | 63.5 | 52.8 |
| Norwegian Nynorsk | M | 61.4 | 53.6 | **61.6** | **63.2** | 60.3 | 53.8 | **60.4** | 51.9 |
| Nuer | VL | 6.9 | 28.7 | 28.3 | 26.1 | - | - | - | - |
| Occitan | L | 63.1 | 61.2 | **65.6** | **65.7** | - | - | - | - |
| Odia (Oriya) | L | 45.8 | 47.6 | **49.3** | 46.1 | - | - | - | - |
| Oromo | VL | 17.1 | 39.1 | **40.0** | 30.4 | 17.2 | 35.4 | 33.6 | 26.9 |
| Pangasinan | VL | 31.3 | 48.5 | 48.3 | 40.7 | - | - | - | - |
| Papiamento | L | 56.2 | 56.1 | **60.9** | **59.4** | - | - | - | - |
| Pashto | L | 36.3 | 38.8 | 35.3 | 33.1 | 33.2 | 36.3 | 33.2 | 27.5 |
| Persian | M | 56.3 | 49.6 | 55.5 | 53.7 | 49.8 | 43.8 | 48.6 | 44.8 |
| Polish | H | 53.1 | 49.0 | 51.9 | 47.6 | 54.6 | 51.5 | 52.5 | 44.0 |
| Portuguese | H | 72.3 | 68.6 | 71.4 | 69.3 | 65.8 | 63.4 | 64.9 | 56.8 |
| Punjabi | M | 48.0 | 48.9 | 48.6 | **50.3** | 44.1 | 48.9 | 45.7 | 46.6 |
| Romanian | M | 65.9 | 60.5 | 64.9 | 63.0 | 60.3 | 55.4 | 58.8 | 54.3 |
| Rundi | VL | 21.4 | 43.9 | 38.4 | 31.7 | - | - | - | - |
| Russian | H | 60.5 | 55.8 | 59.1 | 55.6 | 56.2 | 54.7 | 54.8 | 40.2 |
| Samoan | L | 53.1 | 48.6 | **55.2** | 51.5 | 54.6 | 53.1 | 52.7 | 43.7 |
| Sango | VL | 12.1 | 36.7 | 35.3 | 31.7 | - | - | - | - |
| Sanskrit | L | 33.2 | 28.3 | **36.2** | **34.7** | - | - | - | - |
| Santali | VL | 11.4 | - | **16.8** | **37.7** | - | - | - | - |
| Sardinian | VL | 53.1 | 56.9 | 56.7 | 56.6 | - | - | - | - |
| Scottish Gaelic | L | 54.4 | 50.0 | 53.4 | 50.4 | - | - | - | - |
| Sepedi | L | 37.6 | 51.1 | **54.7** | 48.7 | 35.2 | 37.4 | 35.1 | 31.7 |
| Serbian | M | 61.2 | 57.6 | 60.0 | 61.2 | 46.2 | 44.5 | 44.9 | **51.0** |
| Sesotho | M | 54.5 | 47.9 | **55.2** | 54.0 | - | - | - | - |
| Shan | VL | 2.9 | 39.3 | 33.5 | 34.3 | - | - | - | - |
| Shona | M | 47.1 | 47.8 | 45.9 | 41.1 | 48.2 | 50.1 | 47.1 | 39.7 |
| Sicilian | VL | 46.7 | 42.7 | **51.6** | **46.7** | - | - | - | - |
| Silesian | L | 42.2 | 51.6 | 41.5 | 48.5 | - | - | - | - |
| Sindhi | L | 45.7 | 48.1 | **49.5** | **49.8** | 37.8 | 39.8 | 39.4 | 31.2 |
| Sinhala | L | 53.4 | 45.1 | 50.4 | 51.5 | 50.4 | 44.7 | 47.7 | 45.5 |
| Slovak | M | 62.0 | 57.9 | 60.5 | 59.0 | 60.0 | 56.9 | 57.4 | 50.2 |

| target | resource | FLORES-200 devtest | | | | NTREX | | | |
|---|---|---|---|---|---|---|---|---|---|
| | | PaLM2 S (teacher) | NLLB 1.3B distilled | PaLM2 XXS pt. NTL (mufu20) | Gemma 7B (mufu20) | PaLM2 S (teacher) | NLLB 1.3B distilled | PaLM2 XXS pt. NTL (mufu20) | Gemma 7B (mufu20) |
| Slovenian | M | 58.9 | 54.2 | 56.8 | 54.7 | 58.0 | 53.6 | 55.4 | 54.2 |
| Somali | M | 46.6 | 46.0 | 45.5 | 42.8 | 51.7 | 50.7 | 49.1 | 40.6 |
| Sorani Kurdish | L | 44.3 | 48.7 | 45.0 | 44.5 | 41.5 | 45.3 | 41.1 | 34.6 |
| South Azerbaijani | VL | 28.1 | 26.7 | **35.7** | **32.7** | - | - | - | - |
| South Levantine Arabic | VL | 55.9 | 53.7 | 55.3 | 53.7 | - | - | - | - |
| Spanish | H | 57.2 | 55.2 | 57.1 | 50.4 | 64.9 | 64.1 | 62.7 | 52.3 |
| Sundanese | L | 54.5 | 48.6 | 53.6 | 52.2 | - | - | - | - |
| Swahili | M | 66.0 | 60.0 | 64.6 | 62.8 | 65.7 | 62.7 | 64.6 | 54.3 |
| Swati | VL | 39.6 | 47.0 | 46.4 | 40.6 | 41.0 | 50.2 | 47.4 | 37.6 |
| Swedish | H | 70.6 | 64.8 | 69.3 | 69.8 | 67.0 | 64.1 | 65.8 | 59.1 |
| Ta'izzi-Adeni Arabic | VL | 51.8 | 48.5 | **53.4** | **55.0** | - | - | - | - |
| Taiwanese Mandarin in Traditional script | M | 34.8 | 13.7 | 33.2 | 29.8 | 27.0 | 11.3 | 24.7 | 16.2 |
| Tajik | L | 52.3 | 49.8 | 49.8 | 49.2 | 43.9 | 43.1 | 42.3 | 39.8 |
| Tamasheq | VL | 4.3 | 23.7 | 17.7 | **24.8** | - | - | - | - |
| Tamasheq in Tifinagh script | VL | 6.8 | 17.7 | 17.5 | **27.2** | - | - | - | - |
| Tamazight | VL | 8.4 | 30.4 | 24.3 | **32.2** | - | - | - | - |
| Tamil | M | 59.5 | 56.6 | 57.6 | 58.7 | 48.8 | 48.3 | 47.7 | 47.8 |
| Tatar | L | 48.6 | 48.1 | **50.9** | **49.3** | 45.7 | 48.4 | **49.1** | 42.9 |
| Telugu | M | 59.5 | 56.4 | 57.3 | **59.8** | 46.6 | 45.6 | 45.5 | 39.3 |
| Thai | H | 57.9 | 43.6 | 56.9 | 55.7 | 52.7 | 43.8 | 51.7 | 44.0 |
| Tibetan | L | 32.4 | 34.7 | **39.0** | **36.7** | 28.9 | 33.9 | **36.0** | 30.5 |
| Tigrinya | L | 15.8 | 25.5 | 24.8 | 16.9 | 15.1 | 24.1 | 23.3 | 15.9 |
| Tok Pisin | L | 41.5 | 41.7 | **54.2** | **54.3** | - | - | - | - |
| Tsonga | L | 19.4 | 51.8 | 49.2 | 40.7 | - | - | - | - |
| Tswana | L | 37.9 | 49.3 | 48.3 | 41.2 | 39.8 | 54.5 | 48.2 | 38.3 |
| Tumbuka | VL | 24.3 | 36.3 | **39.9** | 34.9 | - | - | - | - |
| Tunisian Arabic | VL | 45.0 | 40.8 | **47.5** | **48.2** | - | - | - | - |
| Turkish | M | 63.4 | 58.2 | 61.9 | 60.8 | 54.3 | 51.9 | 53.4 | 49.5 |
| Turkmen | L | 49.0 | 41.9 | **53.1** | **50.9** | 43.5 | 38.4 | **44.9** | 40.6 |
| Ukrainian | M | 60.8 | 54.5 | 58.9 | 58.6 | 54.7 | 51.5 | 52.7 | 52.6 |
| Umbundu | VL | 9.8 | 28.0 | 24.2 | **32.0** | - | - | - | - |
| Urdu | M | 48.4 | 48.7 | **49.0** | 46.6 | 50.7 | 50.6 | 50.3 | **51.4** |
| Uyghur | L | 38.6 | 46.4 | 44.0 | 41.0 | 32.4 | 39.9 | 37.9 | 30.8 |
| Uzbek | M | 59.7 | 54.1 | 58.7 | 57.1 | 46.8 | 45.8 | 46.0 | 41.6 |
| Venetian | L | 49.3 | 50.1 | **54.2** | **53.7** | - | - | - | - |
| Vietnamese | M | 61.4 | 57.2 | 60.2 | 59.4 | 61.8 | 59.3 | 60.2 | 57.1 |
| Waray (Philippines) | VL | 55.0 | 56.2 | **64.1** | **62.1** | - | - | - | - |
| Welsh | M | 73.1 | 63.9 | 70.2 | 72.3 | 62.2 | 57.9 | 60.1 | 55.8 |
| Wolof | VL | 14.1 | 27.1 | 25.2 | 27.0 | 15.1 | 30.2 | 26.7 | 24.0 |
| Xhosa | L | 51.7 | 52.7 | 50.0 | 47.7 | 48.7 | 49.2 | 48.0 | 43.6 |
| Yiddish | L | 52.3 | 38.6 | **52.5** | **56.7** | - | - | - | - |
| Yoruba | L | 25.7 | 25.7 | **26.5** | **26.1** | 19.0 | 17.9 | 18.4 | 12.5 |
| Zulu | M | 55.9 | 56.7 | 54.6 | 53.9 | 55.5 | 56.8 | 53.9 | 48.6 |

Table 8: ChrF by 201 language pairs in FLORES-200. VL, L, M and H refer to very-low-, low-, medium- and high-resource languages respectively. Bold values are higher than both the teacher model (PaLM2 S) and NLLB 1.3B.

| | | FLORES-200 devtest | | | | | NTREX | | |
|---|---|---|---|---|---|---|---|---|---|
| | | chrF ↑ (n=201) | chrF ↑ (n=198) | Win% vs. teacher | Win% vs. NLLB 1.3B | Win% vs. NLLB 54B | chrF ↑ (n=112) | Win% vs. teacher | Win% vs. NLLB 1.3B |
| All language pairs | baseline | 28.0 | 28.0 | 21.9 | 2.0 | 0.5 | 23.6 | 5.4 | 0.0 |
| | postedit | 38.6 | 38.7 | 23.4 | 10.6 | 1.5 | 36.8 | 5.4 | 0.9 |
| | mufu5 | 40.6 | 40.7 | 24.9 | 14.1 | 3.5 | 38.5 | 6.2 | 0.9 |
| | mufu10 | **41.0** | **41.1** | 25.4 | 15.2 | 3.5 | **38.9** | 7.1 | 1.8 |
| | mufu20 | 38.8 | 38.9 | 24.4 | 12.6 | 3.0 | 37.1 | 6.2 | 1.8 |
| Low-resource language pairs | baseline | 28.2 | 28.2 | 37.9 | 3.5 | 0.9 | 24.0 | 19.4 | 0.0 |
| | postedit | 33.1 | 33.2 | 40.5 | 6.2 | 1.8 | 30.8 | 19.4 | 0.0 |
| | mufu5 | 35.3 | 35.4 | 43.1 | 8.0 | 4.4 | 31.8 | 22.6 | 0.0 |
| | mufu10 | **35.8** | **35.9** | 44.0 | 9.7 | 4.4 | **32.2** | 25.8 | 0.0 |
| | mufu20 | 33.8 | 33.9 | 42.2 | 8.8 | 3.5 | 31.7 | 22.6 | 3.2 |

Table 9: Mean chrF of BLOOMZ 1B7 finetuned on Mufu, which is analogous to Table 2 in the main text. **Bold** values are the highest chrF scores. Mufu models consistently translate better than baseline and postedit-only.

## A.6 Mufu with BLOOMZ

Using the same Mufu prompts, we finetune BLOOMZ 1B7 and report the mean chrF across language pairs in Table 9.[18] The results corroborate our key findings in the main text, that Mufu-finetuned models are consistently ahead of baseline and postedit-only and achieve the most competitve performance against the teacher in low-resource languages.

## A.7 Mufu self-attention

Tables 10 and 11 are analogous to Table 4, where the attention weights placed over the input by Gemma 2B (mufu5) are highlighted. The examples demonstrate that Mufu models are capable of overriding the postediting target accurately based on semantic alignment across languages beyond orthographic mapping.

## A.8 Failure example: Bad auxiliary input

We identified a few failure cases in Section 4.4 and attribute them partially to poor auxiliary candidates in Mufu input. For example,

> *English: Bird flu, or more formally avian influenza, can infect both birds and mammals.*
> *Automatic Luganda: Enfuba y'enyonyi, oba awamu ey'enfuba y'enyonyi, ey'enyonyi n'ensolo eziyitibwa ennyama.*
> *Automatic Kinyarwanda: Ibirori byamahoro, cyangwa uko byatangiye ibinyamurenge, byatera indwara mu nyamaswa n'ibindi binyabutabire.*
> *Automatic Umbundu: "Otsiku tsiku, tsiku tsiku, tsiku tsiku, tsiku tsiku, tsiku tsiku ...*
> *Automatic Chokwe: Flu wa ndege, nhi cindji cindji cindji cindji cindji cindji cindji ...*
> *Automatic Luba-Lulua: Bu tshisuku tshia nsuku, ni bu tshisuku tshia nsuku tshia nsuku ...*
> *Automatic Lingala: Nzela ya nzoto, to ndenge ya ndenge ya nzoto ya nzoto, ezalaki kozala na nzoto mpe na ndenge ya ndenge ya nzoto.*

Note that Mufu models produce overall worse translations in Lingala than baseline, except for PaLM2 XXS–NTL (Table 8) and PaLM2 XS.

---

[18]BLOOMZ 1B7 model card, see `https://huggingface.co/bigscience/bloomz-1b7`

The English sentence has been translated into Malay, Sundanese, Javanese, Indonesian, Minangkabau and Achinese in Arabic script. These translations may contain errors. Correct the translation from English to Achinese in Arabic script.

English: Imagine, if you will, a Mancunian, Bostonian, Jamaican and Sydneysider sitting around a table having dinner at a restaurant in Toronto.

Automatic Malay: Bayangkan, jika anda mahu, seorang Mancunian, Bostonian, Jamaican dan Sydneysider duduk di sekeliling meja makan di sebuah restoran di Toronto.

Automatic Javanese: Mbayangno, yen sampeyan bakal, Mancunian, Bostonian, Jamaika lan Sydneysider lungguh ngubengi meja mangan nedha bengi ing restoran ing Toronto.

Automatic Sundanese: Bayangkeun, upami anjeun badé, aya Mancunian, Bostonian, Jamaika sareng Sydneysider anu calik di sabudeureun méja tuang di réstoran di Toronto.

Automatic Indonesian: Bayangkan, jika Anda mau, seorang Mancunian, Bostonian, Jamaika dan Sydneysider duduk di sekitar meja makan di sebuah restoran di Toronto.

Automatic Minangkabau: Bayangkan, apobilo indak salah, urang Mancunian, Bostonian, Jamaika jo Sydneysider duduak di sakitar meja makan di restoran di Toronto.

Automatic Achinese in Arabic script: ﻛﻤﻔﻜﻲ, ﻛﻤﻔﻜﻲ, ﻛﻤﻔﻜﻲ, ﻛﻤﻔﻜﻲ, ﻛﻤﻔﻜﻲ, ﻛﻤﻔﻜﻲ, ﻛﻤﻔﻜﻲ, ﻛﻤﻔﻜ , ي , ﻛﻤﻔﻜ

Corrected Achinese in Arabic script: ﻑ ﺍ ﻍ ﺳﻴﻎ , ﻣﻔﭭﻮ ن اﻛ ه , سيدري اورغ منجونيا , باسكتوسيا ,

| reference | ﻧﭬﻴﻜﻴﺮ, ﻣﻔﭭﻮ ﻛﺘﺍ ﺟﺪ, ﺳﻴﺪﺭﻱ ﻣﺎﻧﺠﻮﻧﻴﺍ, ﺑﻮﺳﺘﻮﻧﻴﺍ, ﺟﺎﻣﺎﻳﻜﺍ, ﻏﻦ ﺳﻴﺪﻧﻴﺴﻴﺪﺭ ﻛﺪﻭﻕ ﺑﻚ ﺳﺎﺑﻮﻩ ﻣﻴﺠﺍ ﻛﭬﺎﺟﻮﻩ ﺑﻮ ﻣﺎﻟﻢ ﺑﻚ ﺳﺎﺑﻮﻩ ﺗﻤﻔﺖ ﺳﻤﺎﺟﻮﻩ ﺩ ﺗﻮﺭﻭﻧﺘﻮ. *Nvykyr, mw t Jadu, sydry mnjwny, bwstwny, jmyk, n sydnysydr dwq Bik sbwh myj vjwh B Mlam Bik sbwh tmvt smjwh D twrwntw.* |
|---|---|
| mufu5 | ﭬﺎﻏﺴﻴﻎ, ﻣﻔﭭﻮ ﻧﺎﻛﻪ, ﺳﻴﺪﺭﻱ ﺍﻭﺭﻍ ﻣﻨﺠﻮﻧﻴﺍ, ﺑﺎﺳﻜﺘﻮﺳﻴﺍ, ﺟﺎﻣﺎﻳﻜﺍ ﻏﻦ ﺳﺪﻳﺴﻴﺪﻳﺮ ﺗﺮﭬﻮﻍ ﺑﻚ ﺳﻠﻴﭬﻜﺮ ﻣﻴﺠﺍ ﻣﺎﻛﻦ ﺑﻚ ﺳﺎﺑﻮﻩ ﺭﻳﻨﺴﺘﻮﺭﺍﻥ ﺑﻚ ﺗﻮﺭﻭﻧﻮ *Vsy, mw nkh, sydry awr mnjwny, bsktwsy, jmyk n sdysydr trvw jvw Bik slykr myj mkn Bik sbwh rynstwrn Bik twrwnw.* |
| baseline | ﭬﻴﻜﻴﺮ, ﻣﻔﭭﻮ ﺩﺭﻳﻨﻪ ﺟﺪ, ﺳﺎﺑﻮﻩ ﺍﻭﺭﻍ ﻣﺎﺀﻳﻦ, ﺍﻭﺭﻍ ﺑﻮﺳﺘﻮﻥ, ﺍﻭﺭﻍ ﻛﺎﺍﻣﺎﻥ ﻏﻦ ﺍﻭﺭﻍ ﺳﻴﺪﻧﻴﺴﺍ ﺟﻚ ﺩ ﻛﺮﺟﺍ ﺑﻚ ﺗﻤﻔﺖ ﭬﺎﺟﻮﻩ ﺑﻚ ﺭﻭﻣﻪ ﺗﻮﺭﻭﭬﻨﺲ. *Vykyr, mw drynh Jadu, sbwh awr mayn, awr bwstwn, awr kmn n awr sydnys jk D krj Bik tmvt vjwh Bik Rmah twrwvns.* |

Table 10: Translations from English to Achinese in Arabic script and their romanized form by mufu5 and the baseline Gemma 2B models. جامايكا is correctly transliterated from Jamaika in mufu5, which is attended by the model during its production and is absent in both the postediting target and the baseline translation. Tokens with aggregated attention values under $.02, .06, .14, .24$ are highlighted in white, light gray, dark gray and black respectively.

The English sentence has been translated into Tibetan, Kachin, Bengali, Myanmar (Burmese), Meiteilon (Manipuri) and Mizo. These translations may contain errors. Correct the translation from English to Mizo.

English: At a minimum, you need footwear with suitable soles. Summer shoes are usually very slippery on ice and snow, even some winter boots are deficient.

Automatic Tibetan: ས་མཐའ་ཡང་། ཁྱེད་རང་ལ་ཀུན་པའི་ལྷོག་གི་ཀུན་རིས་དགོས་པ་ཞིག་འདུག དབྱར་གྱི་མགོ་པོ་རྣམས་ག་ང་ར་དང་ཁ་ལ་གཡོལ་མི་སྐྱབ་པ་ཞིག་ཡིན། དེ་བཞིན་དུ་དགུན་གྱི་མགོ་པོ་ལ་ཕས་ཀྱང་དེ་ལྟར་ཡིན།

Automatic Bengali: অন্তত, তোমার উপযুক্ত সোলযুক্ত জুতোর প্রয়োজন। গ্রীষ্মের জুতো সাধারণত বরফ ও তুষারের উপর খুবই পিচ্ছিল, এমনকি কিছু শীতকালীন বুটও অপ্রতুল।

Automatic Kachin: Hku sha sha sha sha sha sha sha sha sha sha sha sha sha sha sha sha sha sha sha sha sha sha sha sha sha sha sha sha sha

Automatic Myanmar (Burmese): အနည်းဆုံးအနေဖြင့် သင်သည် သင်လျော်သော ကြိုးခံနိုင်သည် အောက်ခံများပါဝင်သည် ဖိနပ်များ လိုအပ်ပါသည်။ နွေရာသီဖိနပ်များသည် ရေခဲနှင့် နှင်းများပေါ်တွင် အလွန်ချောလဲလွယ်ပြီး အချို့သော ဆောင်းရာသီဖိနပ်များသည်ပင် ချို့တဲ့နေပါသည်။

Automatic Meiteilon (Manipuri): লৈবাক খৌবাদা, মখোয়না লৈবাক খৌবা লৈবাক খৌবা লৈবাক খৌবা লৈবাক খৌবা লৈবাক খৌবা লৈবাক খৌবা লৈবাক খৌবা লৈবাক

Automatic Mizo: A tlang tlang chuan, kan tlang tlang a rawn tlang tlang chuan, kan tlang tlang a rawn tlang tlang chuan, kan tlang tlang a rawn tlang tlang chuan, kan tlang tlang a rawn tlang tlang chuan.
Corrected Mizo:Thil buaithlak

| | |
|---|---|
| reference | A lo **berah**, I pheikhawk bun chuan kephah siam bik a mamawh a nga.Nipui laia bun thin pheikhawkte hi chu vurah an nal tlangpui a, thlasik laia bun thin pheikhawk thenkhatte pawh hi an la tawk lo cheu a ni. |
| mufu5 | Thil buaithlak **berah** chuan, hotu ropui tak nei ni pe a ngai a, a luahna hotu ring gyhoeddwyd tak te hi tlem leh lus veivah tak te pawh hi a chiang lo a ni. |
| baseline | A tlem chuan, foot hreuh tak hi a thil tih chiang tur a ni. A ver sawh chuan a hmuh a, vur zuah leh vur liah hi a che. |

Table 11: Translations from English to Mizo by mufu5 and the baseline Gemma 2B. The mufu5 model generates *berah* (the most) for the source word "minimum" as it attends to multiple auxiliary translations—some of which are of low quality. The corresponding translations of the word in Bengali (অন্তত) and Myanmar (အနည်းဆုံး) are partially attended to by the model. Tokens with aggregated attention values under .01, .05, .10, .18 are highlighted in white, light gray, dark gray and black respectively.

