# OpenReview forum: "Mufu:  Multilingual Fused Learning for Low-Resource Translation with LLM"
_ICLR.cc/2025/Conference — ICLR 2025 Poster_

### Official Review · Reviewer_uix1 · 2024-10-30

**Soundness:** 3
**Presentation:** 4
**Contribution:** 3
**Rating:** 6
**Confidence:** 5

**Summary:**

This paper introduces Mufu, which turns translation into post-editing task by providing auxiliary translations and target translation from teacher model. The student model learns in-context to produce the correct target translation and is then fine-tuned against references. Languages for auxiliary translations are chosen from URIEL and they evaluate using PaLM S family models along with Gemma 2B,7B on FLORES 200 (iid), and NTREX (ood). The paper contains thorough ablation studies as well as cross lingual attention alignment which helps understanding or interpreting how model is learning through in-context.

**Strengths:**

- The paper is very clearly written and easy to follow.
- They combine 2 interesting learning paradigms: ICL and parameter tuning and their core focus is on very-low and low resource language which I really liked.
- They perform evaluation on NTREX which is important for ood evaluation.
- The experiments performed by authors are quite extensive. I especially liked mufu5hrl, mufu5tr, distilled, and lora which corroborate their approach of selecting 5,10,20 related languages from URIEL.
- Quantitative evidence provided in Figure 3 is quite helpful in knowing how language transfer is taking place. Moreover, the attention pattern further helps in understanding how attention pattern is making mufu models perform better.

**Weaknesses:**

- No model sizes available for PaLM2 family of models. I’m not sure how to compare them with Gemma or NLLB.
- If I were to just compare on the basis of chrF score, only PaLM2 XXS -NLT and PaLM2 S are able to beat NLLB 1.3B distilled model in both FLORES 200 and NTREX (and Gemma 7B on FLORES 200). Rest all are inferior to NLLB 1.3B distilled. One suggestion for authors in this case will be to add `Latency` column for all models (higher for mufu and lower for distilled models) to show the trade off between accuracy and latency which will help readers understand how competitive other models are.
- The authors have mentioned this but finetuning an LLM (or even NLLB with 1B+ param) with just 787 sentences and in-context learning will definitely lead to overfitting which is evident by the fact that mufu20lora performed better than full finetuning. I wonder if that is the case for other models too?
- It’s great they used Gemma 2, an open weight model but I’m slightly disappointed that majority of their experiments use PaLM2 models which are not public like Gemma 2.
- Two iteration process (teacher model followed by student model) is quite expensive. The authors have mentioned that distillation helps to alleviate the problem but it only worked for NTREX in PaLM2 XXS - NTL (not for Gemma 7B), performance on FLORES 200 for both distilled models is lower than NLLB 1.3B.
- The authors experiment with one learning paradigm i.e., in-context learning for LLMs for distillation. Did they try distillation from model outputs (not the one fine-tuned with mufu20)? How much better or worse is in-context learning compared to vanilla distillation?

**Questions:**

- Were there any accidental translations in a different language for Mufu{5,10,20}?
- What exactly is Win% vs teacher? For instance, for NLLB 1.3B distilled, its chrF is 46.0 whereas that of teacher is 43.7, still its win% is 41.3? It means NLLB 1.3B was less than 50% correct when compared to teacher model still its chrF score is higher? Another example, Win% vs teacher is 56.2 for NLLB 54B MoE (48.9 chrF) whereas for mufulora20 with PaLM2 S it is 99% with chrF less than NLLB 54B MoE on FLORES 200. It will be great if authors can formalise what is Win% vs teacher.
- Can the authors explain In theory… model outputs (line 207-211)?

---

> ### Author Response · Authors · 2024-11-16
> **The manuscript is now much improved. We will consider vanilla distillation and ablation with the best performing Gemma model in the final manuscript.**
>
> Thanks for the positive and detailed feedback.
> 1. Unfortunately the sizes of PaLM2 models are not public.  Barring the differences in model size, however, the gap between PaLM2 and Gemma is also driven by differences in their pre-training recipes (e.g., PaLM2 are highly multilingual LLMs, Gemma models are more monolingual); as well as the fact that PaLM2 are relatively older models than Gemma. We speculate PaLM2 XXS to be comparable to NLLB 1.3B due to their similarity in compute requirements. To roughly align these models nonetheless, we suggest comparing their baseline performance in Table 2.
>
> 2. Admittedly, mufu-finetuned models have substantially higher latency than NLLB, which are encoder-decoder models trained specifically for translation. Low-resource languages are however notoriously difficult for LLMs [Robinson et al., 2023; Zhu et al., 2024] due to the huge disparity in resource levels between languages. We now further elaborate this point in lines 500 and 535.
>
> 3. Thanks for pointing out the possibility of overfitting in the other models. Despite seemingly superior performance of mufu20lora based on mean chrF, we find lora to be worse than full finetuning in translations to very-low-resource languages (Figure 2a). In Table 2, PaLM2 XS and PaLM2 S finetuned with the baseline method overfit and perform worse than PaLM2 XXS. This is not the case for Mufu finetuned models, as we see improvement with increasing model capacity. It is also possible that the model overfits to translations in HRLs, but not in LRLs — in which case, a reasonable approach might be to terminate HRL training early (i.e., a form of curriculum learning). We leave this experiment to future work, and now address the point in Section 4.3, line 426.
>
> 4. We conducted most experiments with PaLM2 XXS–NTL as it was the best performing model. Should the work be accepted we will include the corresponding ablation results for the best performing Gemma model in the Appendix.
>
> 5. Thanks for suggesting to compare the distilled mufu20 model against a vanilla distillation setup.  We note, however, that the current teacher model (PaLM2 S) is poor in translations to low-resource languages. We are nonetheless happy to consider other teacher models, and include this result in the Appendix of the final manuscript if required.
>
> 6. Thanks for raising that there might be accidental translations in the wrong language. We manually inspected a number of Indonesian and Chinese languages, and found no absolute incorrect language in the translations. It is however difficult to assess the accuracy systematically as many of these languages share the same scripts, have similar vocabulary, and borrow words from one another (see for example, Table 4). For other language pairs with <20 chrF (e.g., Fon, Tamasheq, Tigrinya), we could only perform sanity checks for accidental translation in the wrong script, as we are unfamiliar with these languages.
>
> 7. Win rate is the percentage of language pairs where the model outperforms a benchmark. For example, NLLB 54B MoE outperforms the teacher in 113/201 ≈ 56.2% language pairs based on chrF; whereas PaLM2 S finetuned with mufu20lora outscores the teacher in 199/201 ≈ 99% language pairs.   We now clarify this in Section 4, line 197.
>
> 8. Thanks for raising that lines 207-211 could be further improved. In Mufu we finetune the models against the gold-standard translations, and expect improvement from the postediting targets. This is effective for LRLs with low-quality postediting candidates. However, improving high-quality translations in HRLs is harder and requires the student model to also learn the subtle differences between model- and human-generated output. It is also possible that the LLM teacher surpasses human for some translations in HRLs, in which case, learning from the human output could be detrimental.  We now elaborate this cause of decline in performance relative to teacher in Section 4, line 263.

---

> > ### Comment · Reviewer_uix1 · 2024-11-22
> >
> > Thank you for your comments, I have some questions from your rebuttal:
> >
> > - You have mentioned that PaLM2 XXS is comparable to NLLB 1.3B, however, performance of PaLM2 XXS (and PaLM2 XS) is lower than NLLB 1.3B. Only when you pretrained it on corpora derived from NTL, some gains were observed.
> > - Thank you for throwing more light on Win% vs teacher. It is now much clear to me.
> > - "However, improving high-quality translations in HRLs is harder and requires the student model to also learn the subtle differences between model- and human-generated output." do you have any specific paper to support your claim?

---

> ### Author Response · Authors · 2024-11-23
>
> Thank you for the questions.
>
> 1. NLLB 1.3B was distilled from NLLB 54B MoE. The latter was trained in hundreds of translation directions, on large-scale mined bitext and monolingual data augmented with backtranslation [1].  Thus it is not surprising that PaLM2 XXS requires further NTL pretraining and gains from Mufu to achieve comparable performance.
>
> 2. Thanks for raising that the statement needs further support. LLMs closely resemble  human translations in their use of lexical and linguistic features [2]. LLM output also becomes increasingly difficult to be distinguished from human translations [3], given the current evaluation method that identifies similar errors in both systems [4]. We now include these citations following the claim in line 265. More importantly, however, we note in the manuscript that the key reason for decline is that finetuning on human-generated translations is suboptimal compared to finetuning on high-quality model-generated data [5].
>
> [1] Costa-jussà, Marta R., et al. "No language left behind: Scaling human-centered machine translation." arXiv preprint arXiv:2207.04672 (2022).
>
> [2] Sizov, Fedor, et al. "Analysing Translation Artifacts: A Comparative Study of LLMs, NMTs, and Human Translations." Proceedings of the Ninth Conference on Machine Translation. 2024.
>
> [3] Kocmi, Tom, et al. "Findings of the WMT24 general machine translation shared task: the LLM era is here but mt is not solved yet." Proceedings of the Ninth Conference on Machine Translation. 2024.
>
> [4] Zhang, Ran, Wei Zhao, and Steffen Eger. "How Good Are LLMs for Literary Translation, Really? Literary Translation Evaluation with Humans and LLMs." arXiv preprint arXiv:2410.18697 (2024).
>
> [5] Finkelstein, Mara, and Markus Freitag. "MBR and QE Finetuning: Training-time Distillation of the Best and Most Expensive Decoding Methods." The Twelfth International Conference on Learning Representations.

---

> > ### Comment · Reviewer_uix1 · 2024-12-02
> >
> > Thank you for your clarification. After carefully going through the paper again, I appreciate the novel approach, but I have some reservations about the latency-accuracy trade-offs. While the concept is innovative, I believe there might be more promising areas to explore, such as developing more distilled models that could potentially address these concerns.
> >
> > At this stage, I'm inclined to keep the scores same. I look forward to potential future improvements in the approach. Thank you and all the best!

---

### Official Review · Reviewer_Xmqh · 2024-11-03

**Soundness:** 3
**Presentation:** 3
**Contribution:** 3
**Rating:** 6
**Confidence:** 3

**Summary:**

This paper tackles low-resource translation quality improvement in LLM models. To maximize data eﬃciency in the low-resource setting, the authors introduce a new approach called Mufu, including automatic selection of multilingual translation candidates and an instruction tuning to correct inaccurate translations via the prompt. Experimental results on Flores-200 dataset for English-XX directions show robustness and achieves better performance against NLLB 1.3B distilled model in 64% of low- and very-low resource language pairs.

**Strengths:**

- experimental results show some effectiveness of the proposed approach
- the idea of leveraging multilinguality via the prompt sounds technically good

**Weaknesses:**

-  unclear about the experimental results; how to decide the best prompt template for mufu; any impacts of language combination used in the prompt template - for example, have you ever tried adding high-resource language translation pairs during training to enhance multilingual training  with high and low-resource language pairs?
-  results are not convincing enough, maybe due to low-resource setting with limited improvement in ChrF. Can you report other metrics such as sacreBLEU scores? Have you tried finetuning LLM with low-resource monolingual data so that the LLM can more effectively enhance Mufu.

**Questions:**

Please see the weaknesses for the questions.

---

> ### Author Response · Authors · 2024-11-16
> **We now include mufu20+5hrl in ablation and report BLEU scores in the appendix.**
>
> Thanks for the review and suggestions.
> 1. We reported the initial prompt selection in Appendix A.1—unfortunately we are unable to move the description to the main text due to page limit. We have tried mufu20+5hrl, which includes 5 HRLs candidates in addition to the same set of auxiliary languages as mufu20, and found it to be less performant than the latter. This result is now reported in Table 2. We are also happy to consider any other suggestions on the mix of auxiliary languages for further ablation studies.
>
> 2. We primarily report our results in chrF rather than BLEU as the latter heavily relies on tokenization that is underdeveloped for many low-resource languages. Nonetheless, we now report BLEU scores in Appendix A.4, which are consistent with the positive results shown in Table 2 in the main text. Prior to mufu finetuning, PaLM2 XXS–NTL has been continued-pretrained on a corpora  derived from Next-Thousand-Language effort (Caswell et al., 2020), which contain monolingual and parallel sentences in 1000+ languages. This results in significant improvement of translation performance when the model is finetuned with mufu prompts, as compared to PaLM2 XXS. Given the gap in performance, we speculate further improvement as well in other models with similar monolingual training.

---

> > ### Comment · Reviewer_Xmqh · 2024-11-22
> >
> > Thank you for addressing my questions and comments. I have updated my score accordingly.

---

### Official Review · Reviewer_84wT · 2024-11-04

**Soundness:** 3
**Presentation:** 3
**Contribution:** 3
**Rating:** 8
**Confidence:** 4

**Summary:**

This paper introduces "Mufu"  , which is a method for low-resource language translation using a multilingual fused learning approach, specifically targeting large language models (LLMs).
The Mufu method, which aims to address the challenge that large language models (LLMs) perform well in translating high-resource languages but still struggle with low-resource languages. The Mufu prompting approach turns the translation task into a post-editing task, leveraging the reasoning capabilities of LLMs with auxiliary translation candidates, requiring the model to assess input quality, align semantics cross-lingually, copy from relevant inputs, and override incorrect instances. Experiments show that LLMs fine-tuned with Mufu-style prompts achieve better performance than the NLLB 1.3B distilled model in 64% of low- and very-low-resource language pairs on the Flores-200 dataset.

**Strengths:**

1. Interesting research, Introduces Mufu, a novel approach leveraging multilingual context and post-editing for low-resource language translation.
2. Employs automatically generated candidates and instructions to correct translations, enhancing LLM's reasoning capability.
3. Demonstrates robustness against poor-quality auxiliary translations, outperforming specialized NMT systems in many low-resource pairs.
4. Proposes a hybrid learning paradigm, combining in-context learning and finetuning for improved translation quality.
5. Implements knowledge distillation to reduce inference costs while maintaining performance gains in low-resource translations.

**Weaknesses:**

1. Experiment Method Optimization， Consider incorporating a more diverse set of low-resource languages in the experimental dataset to better generalize the findings and evaluate the model's performance across a wider linguistic spectrum.

2. Experiment Conclusion Enhancement， Suggest conducting ablation studies to isolate the specific contributions of different components of Mufu, such as the impact of various auxiliary languages, to fine-tune the approach and maximize translation accuracy.

3. 5-shot Prompting Improvement， Explore the use of meta-learning strategies in 5-shot prompting to enhance the model's ability to quickly adapt to new translation tasks with limited examples, potentially improving the efficiency of the learning process.

**Questions:**

1、 more diverse set of low-resource languages in the experimental dataset will be helpful
2、  the impact of various auxiliary languages can be deeply analyzed
3、 prompt analyzation can be improved

---

> ### Author Response · Authors · 2024-11-16
> **We will consider expanding our analysis of multi-head attention to more languages, and improve cross-domain adaptability with 5-shot prompting.**
>
> Thanks for the positive feedback.
>
> 1. Our analyses include 201 languages spoken in 19 regions (e.g., Middle Africa, Central Asia, Northern Europse, etc.), of which 113 languages are considered low- and very-low-resource. While this is only a fraction of 7000+ languages in the world, our setup is unfortunately limited by the availability of high-quality parallel datasets in the other languages.  However, we will consider expanding the set of languages in our analysis of self-attention (Section 4.2), and including in the final draft analyses for other languages analogous to Figure 3 in the Appendix.
>
> 2. We highlighted the contributions of auxiliary languages by comparing mufu5 against mufu5hrl, which consists of 5 HRLs (Dutch, Russian, French, Chinese and Spanish) as auxiliary languages; and mufu5tr, which removes the postediting target in the context. We showed mufu5 to be superior to both mufu5rhl and mufu5tr, indicating the importance of the relevance of auxiliary languages to the target language and  postediting candidates in the context. In Section 4.2, we further corroborated the impact of auxiliary candidates, showing mufu finetuned models to be capable of inferring correct translation from relevant languages by cross-lingual lexical alignment in multihead attention. Some languages are missing in URIEL, we therefore include random auxiliary languages in context. Our preliminary analysis with controlled language resource level show target languages with related auxiliary candidates to have on average higher chrF improvement ratio ((chrF_mufu - chrF_baseline) / chrF_baseline) across models and mufu{5, 10, 20}. We will consider adding in the final manuscript an ablation with the inclusion of random auxiliary candidates for all target languages if required.
>
> 3. Thanks for this suggestion. We will report in the final manuscript the translation performance of the distilled models on NTREX with 5-shot prompting to improve cross-domain adaptability.

---

### Meta-Review · Area_Chair_nnPe · 2024-12-19

**Metareview:**

The submission introduces "Mufu", a method for improving low-resource language translation using a multilingual fused learning approach, tested on large language models (LLMs). The methodology turns translation tasks into post-editing tasks, enhancing translation quality by leveraging the reasoning capabilities of LLMs while using auxiliary translation candidates. Experimental results show that Mufu-style prompts improve translation performance for low- and very-low-resource languages, outperforming NLLB 1.3B distilled model in most language pairs.

Pros:
1. The paper introduces a compelling approach that combines in-context learning and fine-tuning, significantly improving translation performance in low-resource languages, an area that remains a key challenge in the NLP field.
2. The novel methodology of transforming the translation task into a post-editing task enhances the usability and reasoning capabilities of LLMs.
3. The strong experimental results presented across various language pairs demonstrate the robustness of the approach. Additionally, the research provides thorough ablation studies and includes performance evaluations on both in-domain and out-of-domain datasets, making it a substantial contribution to multilingual NLP.


Suggestions:
1. Provide comparative latency metrics across different models to clarify trade-offs between accuracy and efficiency.
2. Consider increasing the diversity of the low-resource languages analyzed to enhance the generalization of findings.
3. Include evaluations using metrics such as sacreBLEU to offer a broader performance perspective.
4. Clarify the size and training details of the models used, particularly for non-public models like PaLM2.
5. Explore alternative distillation techniques beyond the current approach, assessing their effectiveness for LLMs.
6. Delve into overfitting concerns, particularly in high-resource language training, and consider curriculum learning approaches.

**Additional Comments On Reviewer Discussion:**

The reviewers generally agreed on the novelty and technical merits of the paper, with specific praise for its innovative approach and detailed experimental analysis.

Concerns revolved around clarity regarding model details, evaluation metrics, and real-world applicability concerning latency-accuracy trade-offs. The authors responded comprehensively to many queries, committing to additional analyses and clarifications. However, certain reservations about latency and model comparisons were not completely alleviated, leaving room for further exploration and validation.

---

### Decision · Program_Chairs · 2025-01-22

Accept (Poster)